

# New insights on the predisposing factors and geomorphic response to the largest landslide on emerged Earth surface: the Seymareh rock slide - debris avalanche (Zagros Mts., Iran)

Michele Delchiaro[1], Marta Della Seta[1], Salvatore Martino[1], Maryam Dehbozorgi[2], Reza Nozaem[3]

[1]Department of Earth Sciences, Sapienza University of Rome, Rome, Italy
[2]Department of Earth Science, Kharazmi University, Tehran, Iran
[3]School of Geology, College of Science, University of Tehran, Iran

*Correspondence to*: Michele Delchiaro (michele.delchiaro@uniroma1.it)

**Abstract.** The Seymareh landslide, detached ~10 ka from the north-eastern flank of the Kabir-kuh fold (Zagros Mts., Iran), is worldwide recognized as the largest massive rock slope failure (44 Gm$^3$) ever recorded on the emerged Earth surface. Understanding the hazard conditions and the risk associated to this out-of-scale event would provide important pin points for risk mitigation strategies in case of extreme landslide scenarios. Controversial theories have been proposed so far by the scientific community to explain the generation of such an exceptional event and different scenarios have been proposed for explaining the induced changes of landscape. This study provides new constraints to the evolution of the Seymareh river valley, before and after the Seymareh landslide occurrence, to correctly identify the predisposing factors, to suggest possible triggers and deduce the geomorphic response to the slope failure.

We performed detailed geological and geomorphological surveys and mapping of the Seymareh valley and dated with optically stimulated luminescence (OSL) two suites of fluvial terraces (one older and one younger than the Seymareh landslide) as well as a lacustrine terrace (formed after the temporary landslide damming), as useful geomorphic markers of the valley evolution. River profile metrics showed the evidence of a transient landscape and the plano-altimetric distribution of the geomorphic markers has been correlated to the detectable knickpoints along the Seymareh river longitudinal profile. We thus provide time constraints to the main evolutionary stages of the valley before and after the emplacement of the landslide, to be used as inputs for future stress-strain time-dependent numerical modelling in the perspective of calibrating the rock mass viscosity and verifying the possible earthquake trigger of the Seymareh landslide as an ultimate scenario of ongoing mass rock creep processes.

## 1 Introduction

Tectonically active landscapes are very dynamic systems, where threshold conditions on hillslopes are often reached, with considerable implications for natural hazards related to seismicity and to the geomorphic coupling between hillslopes and rivers, with both fluvial control on hillslopes and landslide effects on the fluvial network. In response to rock uplift, relief



and hillslope angles increase linearly in time due to erosional processes in landscapes affected by low to moderate tectonic forcing (Montgomery and Brandon, 2002; Binnie et al., 2007; Larsen and Montgomery, 2012). Nonetheless, such a linear increase in relief and hillslope angles is limited by the reaching of threshold slope conditions associated to the hillslope material strength (Schmidt and Montgomery, 1995), until the latter is exceeded by gravitational stress giving rise to bedrock

landslides. This leads to a nonlinear increase of erosion rates in landscapes affected by long-lasting or high-rate tectonic forcing, where the increase in the rate of channel incision is accommodated by an increased frequency of slope failure rather than by slope steepening.

A vast literature exists on earthquake-induced landslides in tectonically active regions, which is mainly focused on predictive models, based on empirical co-relations derived from databases collected worldwide (Keefer, 1996; Rodriguez et al., 1999;

Li et al. 2004; Owen et al. 2008; Delgado et al., 2011; Hovius et al., 2011; Parker et al., 2011; Jibson and Harp, 2012; Malamud et al., 2014; Martino et al., 2014; Marc et al., 2017), for depicting the expected distribution of the effects. On the other hand, few works (Bozzano et al., 2012, 2016; Della Seta et al., 2017; Martino et al., 2017) have focused on the role of landscape evolution rates on the development of gravitational slope deformations driven by time-dependent rheology, known as Mass Rock Creep (MRC; Chigira, 1992). This kind of deformations can evolve into massive failure due to generalized

slope collapse when the increased strain rate leads to progressive failure associated to strength reduction (Eberhardt et al., 2004; Stead et al., 2006). These collapses lead to rock avalanches originating from the instantaneous fragmentation of rock masses (Hungr et al., 2001) which reach the limit strain conditions when the stationary creep stage evolves in the accelerating creep one. Massive rock slope failures have been widely documented in tectonically active regions and in most cases only speculatively interpreted as earthquake-induced landslides. However, the deformations reached within a rock

mass result from a combination of stress conditions and strain rates, both depending on the shape and dimensions of slopes as well as on the time available for creep evolution. In this perspective, as demonstrated by Bozzano et al. (2016), erosion rates play a key role in the development of MRC processes within the rock masses and, consequently, in their possible evolution into massive rock slope failures even without invoking transient external forcing (e.g. earthquakes).

To infer a more suitable evaluation of the elapsing time for failure in creep evolving slopes, a multi-modelling approach was

recently proposed (Martino et al., 2017), in which Quaternary landscape evolution modelling of slope-to-valley floor systems plays a key role as a tool for chronological constraints to the creep evolution of entire slopes (Bozzano et al., 2016; Della Seta et al., 2017). On one hand, such a modelling can highlight the presence of gravity-induced instability affecting slopes in different evolutionary stages; on the other hand, it allows to reconstruct the timing of the variation of interesting valley sections, thus providing important chronological constraints for the engineering-geological and numerical stress-strain

modelling of slopes, through the plano-altimetric analysis and the dating of geomorphic markers, to be correlated to the anomalies in river longitudinal profiles.

This methodological approach is here applied for the first time to define the evolution of the Seymareh river valley (Lorestan, Iran), before and after this was dammed by the landslide internationally recognized as the largest subaerial rock landslide ever observed. Different evolutionary scenarios were proposed for the Seymareh landslide detached from the anti-





form Kabir-kuh fold of the External Zagros Mountains. Harrison and Falcon (1937, 1938) provided much of the present knowledge on the rock avalanche, including the geology and structure of the source area, the general geomorphology and the basic geometry of the landslide. Oberlander (1965) included a short appendix on the landslide in his study of the Zagros streams and discussed its origin in relation to the activity of the Seymareh River. Later in the 1960s, Watson and Wright

(1969) characterized the geomorphology and stratigraphy of the debris, discussed the origin of the initial rock slide, and examined the debris avalanche emplacement mechanisms. Roberts (2008) and Roberts and Evans (2013) provided new insight into the role of geological factors in defining the original rock slide mass, the detachment mechanisms of the initial rock slide, and the geomechanical behaviour of the rock units involved. Moreover, these papers reported a radiocarbon age that dated back the occurrence of the rock avalanche at 9800 [14]C years BP. Yamani at al. (2012) provided a new

interpretation of the Seymareh landslide emplacement mechanism, in which a sequence of landslide events would be testified by a sequence of entrenched lacustrine terraces upstream of the landslide dam. Finally, Shoaei (2014) reviewed the possible mechanisms of failure and interpreted the post-failure geomorphic features, analyzing the processes responsible for the formation and erosion of the landslide dams of the Seimareh and Jaidar lakes, by using available annual sedimentation data and field measurements of the deposits in these lakes.

This study aims at better understanding the predisposing factors, the geometry, the effects and the hazard conditions associated to the Seymareh landslide, which can be considered as exemplary for providing important pin points to risk mitigation management strategies in case of extreme landslide events. In particular, a revision of the stratigraphic column and some significant geological cross sections have been performed along the Kabir-kuh Fold, to provide new insights on the geo-structural predisposing factors of the landslide. Furthermore, a detailed mapping, plano-altimetric analysis and dating of

the geomorphic markers (Burbank and Anderson, 2012), fluvial and lacustrine terraces pre- and post-failure event, have been performed upstream and downstream of the landslide dam, to better constrain the geomorphological predisposing factors and the geomorphic response to the landslide emplacement.

## 2 Regional geological framework

The Seymareh landslide detached from the north-eastern flank of the Kabir-kuh fold, the longest anticlinal of the Zagros

fold-thrust belt, in the south-western part of Iran (e.g., Sepehr and Cosgrove, 2004). The Zagros chain stretches out from the Tauern Mountains in south-eastern Turkey to south-western Iran, ending near the Strait of Hormuz. It reaches a maximum height of 4548 m in the province of Khuzestan, in the north-western part of Iran, and extends for 2000 km from the Anatolian fault in eastern Turkey (45 ° E, 36 ° E) to the subduction zone of Makran in the south of Iran (26 ° N, 58 ° E) (Mouthereau et al., 2012). The Zagros mountain range is part of the Alpine-Himalayan orogenic system that originates from

the Late-Cretaceous-Cenozoic convergence between Africa/Arabia-Eurasia (Talbot and Alavi, 1996; Stampfli and Borel, 2002; Golonka, 2004; McQuarrie, 2004).



The Zagros were traditionally classified by distinctive lithological units and structural styles into four NW trending tectono-metamorphic and magmatic belts (Fig. 1). These are bounded by defects on a regional scale such as the Main Zagros Thrust (MZT), High Zagros Fault (HZF) and Mountain Front Fault (MFF) (Agard et al., 2005 and references therein). These tectonic units are from the inside to the outside of the belt: the Urumieh Dokhtar volcanic arch, the Sanandaj-Sirjan Zone, the

Imbricate Zone, the Zagros (or Simply) folded belt and the continental Mesopotamian Foreland (Fig. 1). Agard et al. (2005) consider the Main Zagros Thrust (MZT) that separates the Sanandaj-Sirjan area from the Imbricate Zone as the Zagros suture (Fig. 1). The Sanandaj-Sirjan zone represents the accretionary prism of the Arabian margin of the Zagros orogen in which metamorphosed Paleozoic to Mesozoic sedimentary rocks mainly crop out (Mohajjel and Fergusson, 2000 and references therein). Also, calc-alkaline Jurassic to Early Eocene intrusions occur in the tectonic domain. The southwestward boundary

between the Sanandaj-Sirjan zone and the Imbricate zone is defined by the Main Zagros Thrust (MZT). Imbricated tectonic sheets involving radiolarite-ophiolite complexes, Mesozoic and Cenozoic sedimentary and volcanic rocks compose the Imbricate Zone, or High Zagros, which is defined as the innermost deformed part of the Arabian plate. The southwestward boundary between the Imbricate zone and the Simply Folded Belt is defined by the High Zagros Fault (e.g., Agard et al., 2005).

The Seymareh Landslide occurred in the latter tectonic domain, included between the High Zagros Fault (HZF) to the northeast and the Mountain Front Fault (MFF) to the southwest. The Simply Folded Belt involve in spectacular folds the 12–14 km thick sedimentary rocks of the Arabian margin succession covering the continental basement (e.g., McQuarrie, 2004 and references therein). The irregular geometry of the MFF that bounds the Simply Folded Belt southwestward from the Mesopotamian foreland basin, describes salients and reentrants (McQuarrie, 2004; Sepehr and Cosgrove, 2004):

respectively, from northwest to southeast, the Pusht-e Kuh Arc (Lorestan), the Dezful Embayment, the Izeh Zone and the Fars Arc (Fig. 1). A representative balanced cross-section of the Dezful embayment (Blanc et al., 2003) indicates ~49 km of shortening across the Simple Folded Zone. Homke et al., 2004 provide the dates of 8.1 and 7.2 Ma for the onset of the deformation in the front of the Push-e Kush Arc (related to the base of the growth strata observed in the NE flank of the Changuleh syncline) that lasted until 2.5 Ma, around the Pliocene–Pleistocene boundary. A long-term shortening rate of ~10

mm y$^{-1}$ was derived for the deformation in the Simple Folded Zone, which is the same as the present-day one derived by GPS measurements (Tatar et al. 2002).

Seismicity is distributed in a 200-300 km wide area of the Zagros mountain range, with a sharp cut along the Main Zagros Reverse Fault in NE. Looking at the depth and magnitude of recent earthquakes (Fig. 2), the seismogenic faults can generate recurrent earthquakes of Mw 5-6 and exceptional earthquakes of higher magnitude, i.e. up to Mw 6-8. These seismogenic

faults follow the general trend of the Zagros, having NW-SE direction in the northwestern portion of the chain, while in the southeastern part they assume an E-W trend; they are characterized by high-angle planes (40-50°) reaching depths between 4 and 19 km (Hatzfeld et al., 2010; Paul et al., 2010, Rajabi et al., 2011). The earthquakes, which originate at a variable depth of 12-19 km, are probably located in the crystalline basement or at the interface with the Cambrian-Pliocene cover, whose thickness reaches about 12 km. The shallowest earthquakes, located at 4-8 km of depth, are located inside the sedimentary



cover and, in general, these events do not produce surface ruptures, probably due to the presence of marly and evaporitic levels that accommodate the deformation (Hatzfeld et al., 2010; Leturmy and Robin, 2010; Navabpour et al., 2010; Paul et al., 2010; Saura et al., 2011).

The Seymareh landslide occurred in a very densely seismic area, so that Roberts and Evans (2013) hypothesized that seismic forcing may have played a primary role in triggering the landslide.

## 3 Geomorphological background

The Zagros Range globally provides one of the most spectacular examples of landscape evolution in response to active tectonics (Borne and Twidale, 2011), since its drainage network clearly adapted to the growth of the thrust-fold structures, also in relation with the erodibility of the outcropping formations (Oberlander, 1985).

Several landscape evolution models have been carried out to explain the drainage history of the Zagros in association with tectonic deformation of the area. One of the most famous models was suggested by Oberlander (1968, 1985), who interpreted the transverse rivers cutting the structural highs NE of Dezful as a consequence of the regional stratigraphy. His observations focused on the 'tangs' of the Dezful Zagros. The tangs are steep, narrow gorges cut by rivers transverse to the fold length, often through the highest structural and topographic part of the anticline crests. Oberlander suggested that the

morphology of the gorges varied from V-shaped valleys in the older northern parts of the mountain belt, to slot-like canyons in the Simple Folded Belt, the present-day focus of deformation. He believed that the pattern was repeated too often to be by chance, and that previous explanation for transverse drainage formation (e.g. that the rivers are antecedent to the structure) did not account for the close proximity of the tangs to the anticline structural highs. Oberlander (1968) suggested that the drainage network in the NW Zagros was instead superimposed from structurally conformable younger horizons. In his

model, the breaching of hard geological units of antiformal ridges follows a phase of river cutting and expansion of the fold axial basins through the softer overlying units. This is cyclical, according to the alternating "mobile" and "competent" units in the Zagros fold stratigraphy. In the Kabir-kuh fold of the simply folded belt of Lorestan, the transverse cutting of the Asmari limestone, and the exposure of the underlying easily erodible Pabdeh-Gurpi marls, results in the formation of a low-relief landscape with synformal ridges on which new through-going drainage system can be developed. These new drainage

systems are then superimposed on the Mesozoic Bangestan limestone as it is exhumed by continued fold growth. In Oberlander's hypothesis, it is the Pabdeh and Gurpi marls that facilitate the creation of a low-relief landscape across the anticline crests and are therefore integral to the story of drainage superimposition. However, the deep-marine Pabdeh marls, which are sandwiched between the Asmari and Bangestan limestones, grade southeastward into the dolomitic, sabkha-type Jahrum limestone (Sepehr and Cosgrove, 2004; Sherkati et al., 2005). As such, a significant thickness of soft, easily erodible

sediment is not present in Fars province, where the development of a low-relief landscape on which to generate new through-going drainage systems evolves in response to folding in hard and resistant lithologies. Lateral variations in stratigraphy are thus shown to exert a major control on the patterns of drainage, and hence sedimentation, at a regional scale. It is likely that,



moving westwards through the Zagros, the gradually increasing thickness of the soft Pabdeh marls can be the cause for a gradual transition towards the type of landscape described by Oberlander (1968, 1985) in Dezful.

Ramsey et al. (2008) proposed an alternative model that considers the response of drainage network to active folding. They suggested that the rivers are diverted around the tips of laterally growing anticlines until the merger of individual fold segments becomes important in forcing drainage evolution. Fold segmentation, lateral growth and linkage has profound importance on the development of through-going drainage and sedimentation patterns in the Fars province of the Zagros. In the absence of independent constrains on the timing of landscape evolution, Ramsey et al. (2008) described the fan-shaped tributary patterns on fold flanks and other geomorphological features (such as wind gaps and water gaps, deflection of rivers parallel to anticline axes and trapping of rivers between fold tips) as indicators of drainage evolution in the present-day landscape.

In the less uplifted areas of the chain, the initial stages of cross-river evolution are still ongoing, and this makes it possible to formulate ergodic models for landscape evolution, also in relation to the fundamental role that stratigraphy plays in the development of transverse drainage in uplifting chains. Tucker and Slingerland (1996) computed a numerical landscape evolution model, calibrated on the Kabir-kuh fold, to understand how the growth and propagation of the folds, the different lithologies and the drainage network can influence the sediment flux from a tectonically active belt towards the foreland basin. The authors calibrated the landscape evolution model with the current topography of the range, obtaining time constraints for landscape evolution modeling. According to the Oberlander model, Fig. 3 shows four main steps to describe the landscape evolution of the Kabir-kuh fold with the timing provided in the model by Tucker and Slingerland (1996).

Step 1 - About 4.3 Ma, in response to the initial stages of fold growth, an orthoclinal drainage develops, parallel to the main structures. The tributaries flowing along the flanks of the folds transport debris which is deposited in the synclines. In the Kabir-kuh fold the carbonate core is still buried by the Miocene cover units.

Step 2 - About 3.8 Ma, as soon as the deformation front migrates towards SW, new folds raise with a progressive adjustment of the drainage to these morpho-structures. The previously deposited sediments are remobilized and transported towards the depocenter of the syncline basins and partly outside; the syn-orogenic deposits are strongly eroded along the crests of the anticlines, thus exposing the underlying formations. This causes a topography characterized by resistant hogbacks that border the inner cores.

Step 3 - About 2.4 Ma, with the ongoing of deformation, the drainage develops in a "trellis" pattern. The river erosion affects the erodible units located stratigraphically between the limestone of Asmari formation and the inner core of the fold. At the end of this step the Miocene cover is completely removed from the ridges and the river erosion also affects the marls and evaporites of the syn-orogenic formations in the valleys, exposing the underlying limestone of the Asmari Fm.

Step 4 - About 1.6 Ma, due to the continuous uplift and exhumation of younger, more external folds, the sediment accumulation becomes negligible and the Asmari limestone is strongly eroded giving rise to syncline ridges. The



following Quaternary landscape evolution is then likely driven by the evolution of the drainage network, also influenced by climatic factors, and by the slope-to-channel dynamics.

The model by Tucker and Slingerland (1996) is the unique numerical model existing on the Kabir-kuh fold and this motivates our choice of using it as a reference for the medium-to-long term evolution of the Seymareh river valley. The

Seymareh river valley is arranged parallel Kabir-kuh fold and its evolution was inevitably influenced by the exceptional landslide event that temporarily dammed it, causing the formation of three lakes (Fig. 4). The valley evolution before and after the event is well recorded by Quaternary landforms preserved along the valley. Yamani et al. (2012) focused on the post-failure evolution of the valley and interpreted four levels of terraces upstream the landslide dam as lacustrine terraces formed in response to a sequence of landslide events. Shoaei (2014), in addition to evaluating the longevity of the Seymareh

landslide dams, identified in the merging of Seymareh River with a left tributary the reason for strong river incision at the base of the north-eastern flank of the Kabir-kuh fold as possible causal factor for the Seymareh landslide collapse.

However, none of the previous study on the Quaternary evolution of the Seymareh river valley neither provided absolute dating of the geomorphic markers (mainly fluvial terraces) preserved upstream but also downstream of the landslide dam, nor provided robust and quantitative constraints to the pre-failure valley evolution as possible geomorphological factor for

failure occurrence.

## 4 Revised stratigraphic column and geological sections of Seymareh river valley

The Seymareh Landslide detached from the north-eastern flank of the anti-form Kabir-kuh fold into the Simply folded Zagros in the Lorestan region. The outcropping formations in the anticline area refer to a time interval ranging from the Upper Cretaceous to the Lower Miocene and are characterized by different lithological and rheological properties (Vergés et

al., 2011). In this regard, Tucker and Slingerland (1996) referred to a 300-m-thick resistant unit, corresponding to the Oligocene Asmari Limestone over a 1000-m-thick weak unit corresponding to Campanian to Eocene flysch (Pabdeh-Gurpi Fm.), while, Roberts and Evans (2013) referred to an involved section including: 225 m of limestone of the Asmari Fm., 525 m of marls of the Pabdeh Fm. and the upper part (50 m), mainly marls, of the Gurpi Fm. Since the geo-structural setting of the fold flanks could have been a crucial predisposing factor for the catastrophic massive rock slope failure, we revised the

stratigraphic column and performed some geological sections of the Seymareh valley. Specifically, the investigated area includes the middle and low reaches of Seymareh River starting ~ 60 km upstream of the Seymareh landslide down to the SE termination of the Kabir-kuh fold. In Fig. 5 the geological map of the Seymareh river valley is reported. At the base of the stratigraphic column there is the Sarvak Fm. (Cretaceous, thickness > 750 m) consisting of a thick carbonate unit that represents one of the largest reservoirs for hydrocarbons in Iran (Elyasi et al., 2014). At the top of the Sarvak Fm. there is the

Ilam-Surgah Fm. (Upper Cretaceous, thickness about 250 m), consisting of limestone of transgressive-regressive foredeep facies deposited in the pro-foreland basin (Elyasi et al., 2014). The Ilam-Surgah Fm. is limited at the top by Gurpi Fm. (Upper Cretaceous, thickness about 400 m) consisting of a marly limestone, marl and hemipelagic shales of deep marine



facies associated to the progressive migration towards S of the pro-foreland areas, which are in unconformity with the Sarvak and in onlap with the Ilam-Surgah (Elyasi and Goshtasbi, 2015). Within the Gurpi Fm. it is possible to recognize a considerably more calcareous horizon called Emam Hassan Member (25 m thick). Above the Gurpi Fm. there is the Pabdeh Fm. (Upper Paleocene - Lower Oligocene, thickness about 350 m) consisting of hemipelagic-pelagic calcareous shales

(Elyasi and Goshtasbi, 2015). Specifically, a detailed study on this formation was performed on the stratigraphy outcropping along the north-eastern flank of the Kabir-kuh fold, thus allowing the discrimination of several lithostratigraphic-based members. The Pabdeh Fm. is composed of three members: 1) the lower Pabdeh member (150 m thick), which is dominated by marls and shales, 2) the Taleh Zang member (50 m thick), consisting of platform limestone, and 3) the upper Pabdeh member (150 m thick), composed mainly of calcareous marl. The succession is completed by the Asmari Fm. (Oligocene –

Miocene, thickness about 200 m), which creates a carbonatic carapace originally covering the top of the Kabir-kuh fold. The Asmari Fm. consists of alternating fossiliferous, massive, thinly stratified gray-brown limestone, microcrystalline limestone, dolomitic limestone, and marly limestone (Khoshboresh, 2013). In the synclinal valleys between the Kabir-kuh fold and the adjacent ones, the Asmari Fm. is overlapped by a Miocene-Pliocene succession (Homke et al., 2004). Referring to the Changuleh syncline studied by Homke et al., 2004, the latter foreland stratigraphy include: i) the Gachsaran Fm. (Late Lower

Miocene - 12.3 Ma, thickness about 400 m), composed of salt, anhydrite, marl and gypsum; ii) the Agha Jari Fm. (12.3 Ma – 3 Ma,, thickness about 1400 m); and iii) the Bakhtiari Fm. (3 Ma – Early Pleistocene, thickness about 900 m). More in particular, the Gachsaran Fm. consists of sandstones and conglomerates, linked to the evolution from deltaic to fluvial transitional environments (Elyasi et al., 2014) while the Bakhtiari formation consists of conglomerates characterized by coarse and mud-supported grains, sandstones, shales and silts and marks the onset of syn-orogenic fluvial environment

conditions (Shafiei and Dusseault, 2008). Figure 5 shows also the revised stratigraphic column and three geological cross-sections related to different structural sectors are reported.

All the reported cross-sections intersect the synclinal valley of the Seymareh River. Specifically, the dip angle of the syncline north-eastern flank considerably decreases from NW to SE from 45° (section A-A') to 18-20° (section B-B'), down to 5-15° (section C-C'). As a consequence, along the section A-A' the Cretaceous-Palaeocenic bedrock (from the Sarvak Fm.

to the Asmari Fm.) offers a greater accommodation volume to the continental and epicontinental formations (Gachsaran Fm. and Agha Jari Fm.), since the synclinal axis is located at a lower elevation than in B-B 'and C-C' sections. For this reason, in this sector of the valley it is possible to observe extensive outcrops of the Agha Jari Fm.

From the topographic point of view, the section crossing the Seymareh landslide (B-B') shows about 1600 m of relief between the highest elevation of the carbonatic carapace outcrop of the reconstructed Asmari Fm. (about 2100 m a.s.l.) and

the river bed elevation (449 m a.s.l.), which is considerably higher if compared to section A-A' (820 m) and C-C' (1180 m).

Moreover, along section A-A', the Seymareh river bed does not coincide with the syncline axis, as it is located northeastwards. In this regard, along this sector of the valley, terraced conglomerate deposits crop out extensively. Their geometry (section A-A' in Fig. 5) can be associated with alluvial fans generated on the flanks of a former synclinal valley, by streams likely being the tributaries of a Paleo-Seymareh River whose path was to the SW of the present one. The absence



of a kinematic release at the base of the hillslope and of a connectivity between the slope and valley floor evolutions imply that this sector is characterized by sliding phenomena mostly controlled by the structural arrangement of the flatiron slopes of the north-eastern flank of the Kabir-kuh. The slope-to-valley floor interaction is evident in the B-B' and C-C' sections, where the river incised the carbonate caprock (Asmari Fm., section B-B') and, led to the formation of a canyon about 200 m deep in the Asmari Fm., although the latter is not yet completely released (section C-C').

## 5 Evolutionary analysis of the Seymareh valley

### 5.1 Methods

The geomorphological study of the area was carried out firstly through the analysis and interpretation of remote data (aerial photos and Google Earth satellite optical images) vector topographic maps (scale 1:25,000), which led to the first detection of possible geomorphic markers within the Seymareh river valley, to the construction of a 10m Digital Elevation Model (DEM) for terrain analyses, and to the projection of the possible geomorphic markers along the river longitudinal profile. The DEM was obtained by ArcGis 10 software package, starting from vector topographic data derived from literature and using the ANUDEM interpolation algorithm (Hutchinson et al., 2011 and references therein). A geomorphological field survey was then carried out with the aim of mapping the most significant active and relict landforms for the Quaternary evolution of the Seymareh river valley and sampling the corresponding lacustrine and fluvial deposits in order to date them with the OSL method (Optically Stimulated Luminescence; Murray and Olley, 2002; Wintle and Murray, 2006 and references therein). For each site, we sampled at a depth >1 m below the top depositional surface or the eventual erosional surfaces recognized within the deposit to avoid the risk of rejuvenated ages. The OSL data where acquired at the LABER OSL Laboratory, Waterville, Ohio (U.S.). Quartz was extracted for equivalent dose (De) measurements. In the OSL laboratory, the sample was treated firstly with 10% HCl and 30% $H_2O_2$ to remove organic materials and carbonates, respectively. After grain size separation, the fraction of 90-125 μm is relatively abundant, so this fraction was chosen for De determination. The grains were treated with HF acid (40%) for about 40 min to remove alpha dosed surface, followed by 10% HCl acid to remove fluoride precipitates. Luminescence measurements were performed using an automated Risø TL/OSL-20 reader. Stimulation was carried out by a blue LED (λ=470±20 nm) stimulation source for 40 s at 130°C. Irradiation was carried out using a 90Sr/90Y beta source built into the reader. The OSL signal was detected by a 9235QA photomultiplier tube through a U-340 filter with 7.5 mm thickness. For De determination, SAR protocol was adopted. The preheat temperature was chosen to be 260°C for 10 s and cut-heat is 220°C for 10 s. The final De is the average of Des of all aliquots, and the error of the final De is the standard error of the De distribution. For each sample, at least 12 aliquots were measured for De determination. The De was measured using SAR on quartz, and the aliquots that passed criteria checks were used for final De calculation.

Recycling ratios were between 0.90-1.1. Recuperation is relatively small. The cosmic ray dose rate was estimated for each sample as a function of depth, altitude and geomagnetic latitude. The concentration of U, Th and K was measured by neutral



activation analysis (NAA). The elemental concentrations were then converted into annual dose rate, taking into account of the water content (lab measured) effect. The final OSL age is then: De/Dose-rate.

In order to automatically extract the hydrographic network from DEM and then to project the geomorphic markers along the longitudinal river profiles, some of the ArcGIS 10 tools of the Hydrology toolbox were used, setting the flow accumulation

threshold according to that proposed for the fluvial domain ($10^{-1}$ km$^2$) by Montgomery and Foufoula-Georgiu (1993). The longitudinal profile was therefore transformed into a route along which the elevation of the top surfaces of geomorphic markers identified in the area were projected through the Linear Referencing Tools.

**5.2 Results**

The best geomorphic markers preserved in the study area are represented by a lacustrine terrace and two suites of fluvial

terraces, which mark to the evolutionary stages of the valley, respectively before and after the landslide emplacement. These markers are reported in the following list.

• In the middle reach of the Seymareh river valley, conglomerates (Cg_m) pertaining to inactive, terraced alluvial fans and a suite of four orders of fill terraces (named from Qt1_m to Qt4_m). The latter are entrenched in the terraced lacustrine deposit of Seymareh Lake upstream of the landslide, in the area where Harrison and Falcon (1938), Roberts and Evans

(2013) and Shoaei (2014) hypothesized the natural damming lake could be extended (Figs. 6 and 7). Here we dated successfully 4 samples (SEY4, SEY5, SEY6, SEY8; Table 1 and supplementary material).

• A suite of 2 strath terraces and a flood plain shaped onto the landslide debris along the Seymareh river gorge (Figs. 8 and 10). Here we dated successfully one sample taken on a strath terrace (SEY9; Table 1 and supplementary material).

• In the lower reach of the Seymareh river valley, conglomerates (Cg_l) pertaining to inactive, terraced alluvial fans and a

suite of four orders of fill terraces (named from Qt1_l to Qt4_l) downstream of the Seymareh landslide (Figs. 9 and 10). Here we dated successfully three samples from the fill terraces deposits (SEY3, SEY10, SEY11; Table 1 and supplementary material).

Fluvial and lacustrine terrace deposits mainly consist of gravel, sand, silt and clay, while conglomerates outcropping immediately upstream and downstream of the landslide pertain to inactive alluvial fans connected to a relict position of the

valley floor, likely of a Paleo-Seymareh River. The strath terraces and flood plain developed onto the landslide debris are important markers of the evolution of the natural dam, since they formed after its cut likely due to an overflow of the damming lake.

The above described geomorphic markers of the Seymareh river valley have been mapped and reported in morpho-stratigraphic profiles. The landforms most significant for the valley slopes evolution are here presented with a detail for the

post-failure fluvial and lacustrine terrace suites upstream of the landslide dam (Fig. 7) and the pre-failure fluvial terrace suite downstream of the landslide dam (Fig. 10), respectively.

Figures 11 and 12 report the longitudinal profile of Seymareh River, along which were projected besides the geomorphic markers:



a) the benchmarks of the basal contact of the Quaternary deposits on the bedrock;

b) the projection of points corresponding to the top of the Seymareh landslide debris;

c) the upstream and downstream limits of the landslide;

d) the location of the OSL sampling;

e) the projection of the outcrop of the Bakhtiari Fm. (Fig. 13), which is rarely preserved and marks the initial alluvial infill of the Seymareh valley.

Figure 11 shows the height distribution of the pre-failure geomorphic markers. The benchmarks along the Seymareh River indicate a mostly bedrock channel and the longitudinal profile is characterized by two knickpoints respectively located upstream of the Seymareh landslide and downstream of the lowest suite of alluvial terraces, as indicated by the black arrows.

The major knickpoint is located immediately upstream of the Seymareh landslide and is the most interesting to be analyzed with respect to the landslide event. Its shape in the long profile clearly let us identify it as a "slope-break knickpoint" (Kirby and Whipple, 2012; Boulton et al., 2014), thus developed as a knickpoint migrating in response to a persistent perturbation to the fluvial system (Tucker and Whipple, 2002), as frequently observed in tectonically active regions. Figures 11 and 12 show that the preserved geomorphic markers do not belong to the same suite of terraces, as their projections along the Seymareh

River do not have each other any topographic correlation. The top of all the fluvial terraces downstream of the Seymareh landslide is located lower in height than the most important knickpoint located immediately upstream and sculpted in the bedrock. The location of this knickpoint upstream of the Seymareh landslide and the outcrop of the basal contact of the landslide at the bottom of the Seymareh gorge (Fig. 8a) testify that this shape of the longitudinal profile was already developed before the failure, that means that the erosion wave which generated the knickpoint affected the Seymareh

landslide slope foot before the occurrence of failure.

The poorly preserved, well cemented alluvial fan conglomeratic deposits outcropping upstream of the landslide lie on the Miocene Agha Jari Fm., at elevation higher that the outcrops of the Bakhtiari Fm. Their remnants are aligned in correspondence of the axis of a relict synclinal valley, likely corresponding to a very early stage (Pliocene?) of the Seymareh valley evolution.

On the other hand, the conglomerate deposits outcropping downstream of the landslide (Cg_l) are closer in height to the major knickpoint, thus suggesting that they were in equilibrium with a local base level corresponding to the early propagation of the major knickpoint. Furthermore, they must be younger than the Bakhtiari Fm., which is preserved at higher elevation.

The alluvial terraces located downstream of the Seymareh landslide likely mark the valley evolutionary stages during the

major knickpoint retreat (Demoulin et al., 2017). Unfortunately, along the longitudinal river profile the uppermost outcrops of each level of this terrace suite were swept away by the landslide. Nonetheless, the deposits of the three youngest levels downstream of the landslide were suitable for OSL dating (samples SEY3, SEY10 and SEY11, respectively) and provided useful time constraints to the main depositional events during the knickpoint retreat. Minimum ages of 373±34 ka and 312±45 ka have been obtained for samples SEY3 and SEY10 respectively since this samples were saturated due to their very



little quartz grains), while SEY11 was dated at 60±5 ka. According to what observed by Bridgland et al. (2017) about river terrace development in the NE Mediterranean region, the sedimentation phases should correspond to cold periods. In particular, Bridgland et al. (2012) observed, in the valleys of the Tigris and Ceyhan in Turkey, the Kebir in Syria and the trans-border rivers Orontes and Euphrates, a regular terrace formation in synchrony with 100 ka climatic cycles to be

correlated with MIS 12, 10, 8, 6 and 4-2. Therefore, the minimum ages obtained for the SEY3 and SEY10 samples could be reasonably extended to 478 ka (MIS 12) and 374 ka (MIS 10), respectively, while the OSL age of the SEY11 fits well with the Last Glacial Period.

Figure 12 shows the height distribution of the post-failure geomorphic markers. They are represented by: i) a horizontal lacustrine terrace formed by the incision of the deposits pertaining to the Seymareh lake, formed as a consequence of the

landslide damming; ii) two levels of strath terraces and a flood plain formed on the landslide debris during the initial stages of dam cutting and emptying of the lake; iii) four levels of fill terraces formed after the emptying of the Seymareh lake. We sampled and dated the lacustrine deposit at two different stratigraphic levels, at 560 and 590 m a.s.l., which provided OSL ages of 10.4±0.90 ka (sample SEY8) and 7.37±0.73 ka (sample SEY4), respectively. The OSL age of 17.9±1.50 ka (SEY6) obtained for an alluvial deposit at the base of the lacustrine deposits is in coherent with the age of emplacement of the

Seymareh landslide, as already inferred by Roberts and Evans (2013).

Despite their interpretation as progressively younger lacustrine deposits by Yamani et al. (2012), the four levels of terraces entrenched in the lacustrine deposit show a longitudinal downstream gradient, which, along with their sedimentological characters, identify them as fill terraces. Furthermore, the OSL age obtained for the lacustrine deposit at the base of the Qt2_m terrace (sample SEY8) is 10.4±0.90 ka, testifying that the suite of alluvial terraces is all entrenched into the same

(and unique) lacustrine deposit. The OSL age of 4.49±0.48 ka obtained for the Qt1_m terrace (sample SEY5) provides time constraints to the emptying phase of the Seymareh lake. Such time constraints are fine-tuned by the age of the strath terrace formed on the landslide debris, which testify to the initial stage of lake emptying at 6.59±0.49 ka (SEY9).

## 6 Discussion

The here obtained results provide significant time constraints for the evolutionary model of the Seymareh valley before and

after the natural damming caused by the Seymareh landslide. This giant event was already dated to ~9.8 ka by Roberts and Evans (2013). This study provides new insights about the predisposing factors and the geomorphic response of the valley system to such a catastrophic phenomenon. A first consideration derives from the geological succession outcropping on the north-eastern slope of the Kabir-kuh, since the layering of formations (Fig. 13), which could be associated to different rheological behaviors, could induce differential strain rates within the slope which can justify the strain evolution toward

failure according to a MRC process. More in particular, the time-dependent visco-plastic behavior, more typical of clayey and marly deposits, which have lower viscosity values, justifies time-dependent (creep) strains which generate high stress concentration within the higher viscosity level over time (i.e. mostly characterized by elasto-plastic rheology), inducing their





cracking and leading to failure mechanisms. Clear evidences of these rheological effects have been recognized in gravity-induced folding within the thin-layered Pabdeh Fm. as well as in impressive buckling of its downslope dipping strata which crop out just along the sliding surface of the Seymareh landslide (Figs. 14b and 14c). As for the kinematic constraints, the geological setting and the attitude of layers in addition with the slope dip represent strong predisposing factors for a rock

mass sliding mechanism. In fact, along the NE flank of the Kabir-kuh fold, the outcropping succession is characterized by a stiffness contrast between the upper member of the Pabdeh Fm. and the overlying Asmari Fm. The attitude of strata is likely moderately dipping downslope (15°-20°), and a reduced lateral confining effect is due to continental and epi-continental deposits ascribable to Gachsaran and Agha Jari Fms. Moreover, the lower dip angle reduced the vertical thickness of the carbonate Asmari Fm. caprock which was completely eroded by the Seymareh River during its engraving, thus causing the

kinematic release at the slope toe. Moreover, the topographic relief of 1600 m between the highest outcropping point of the carbonate formation and the lowest point of the sector where the landslide occurred is significant for the potential energy of the slope mass in deformation and much higher than in the adjacent sectors of the Kabir-kuh ridge. Finally, a relevant kinematic freedom degree was created by the engraved network of gullies that dissect the Asmari Fm. carbonate caprock. The stress release at the bottom of the slope was likely produced by the Middle-Late Pleistocene upstream migration of the

knickpoint along the Seymareh river longitudinal profile. Unfortunately, since the emplacement of the landslide swept away the uppermost outcrops of the alluvial terraces formed in response to the knickpoint upstream migration, the rate of knickpoint migration cannot be inferred. Nonetheless, an elapsing time to failure in the order of $10^2$ ky can be reasonably attributed to the slope-to-valley floor system before the generalized failure occurrence.

The geomorphic response of the valley to the 44 Gm³ natural dam was the formation of three lakes (Seymareh, Jaidar and

Balmak; Fig. 4) whose persistence and evolution is well recorded by the deposits outcropping in the valley. In this regard, the estimation of a sedimentation rate of 10 mm y⁻¹ in the Seymareh Lake was obtained using the OSL ages of 10.4±0.90 ka and 7.37±0.73 ka for the lacustrine deposit sampled at 560 and 590 m a.s.l., respectively. Furthermore, the strath terrace sculpted on the landslide deposit and dated at 6.59±0.49 ka constrains the cut of the natural dam due to overflow which caused the progressive lake emptying. The lake overflow was likely caused by the progressive filling of the reservoir with

lacustrine deposits, which reduced progressively the dam infiltration section. Nevertheless, the possible role of groundwater seepage within the pervious natural dam in balancing the Seymareh river discharge and delaying the dam overflow remains a questionable topic to be approached and solved in future studies. As it results by the age of the Qt1_m terrace (of 4.49±0.48 ka), the Seymareh lake persisted likely up to ~5 ka. Since the top of the lacustrine deposit lies at 630 m a.s.l., an increased sedimentation rate of ~17 mm y⁻¹ can be inferred for the late stage of the lake evolution, which is in agreement with an

increased sediment yield from tributaries during the early stages of lake emptying (Fig. 15).

The overflow, at 6.59±0.49 ka allows to calculate the erosion rate affecting the landslide deposit after the overflow. The ratio between the thickness of the eroded sediment (~120 m) and the time elapsed since the beginning of the process (~ 6.59 ky) allows to estimate an erosion rate of 1.8 cm y⁻¹ for the Seymareh River along the gorge. The cut of the landslide dam induced a new change in the fluvial base level, bringing the slope-to-valley floor system into disequilibrium. For this reason, a dense



drainage system was set on the scar area which, thanks to the high erodibility and low permeability of the less competent Pabdeh-Gurpi Fm. immediately below the sliding surface on the Kabir-kuh NE slope, has generated the badlands mapped in Fig. 7. The time scan of landscape evolution in the Seymareh river valley before and after the failure occurrence can be summarized in the following listed six phases:

1. Setting of a Paleo-Seymareh River into a synclinal valley, likely developed in the Pliocene, to the west of the present position of the Seymareh River and deposition of fan deposits (Cg_m) (Fig. 16a).

2. Development of the valley with local base level correlated to the Seymareh longitudinal profile segment upstream of the major knickpoint along the Seymareh River and coeval to the deposition of the Bakhtiari Fm. (Late Pliocene-Early Pleistocene) (Fig. 16b).

3. Emplacement of the downstream fan deposits corresponding to the Cg_l conglomerates (Early Pleistocene) and generation of the four orders of Middle-Late Pleistocene alluvial terraces (Qt1_l-Qt4_l) preserved downstream of the landslide and formed during the progressive migration of the major knickpoint, which is presently located upstream of the landslide (Fig. 16c).

4. Seymareh landslide event (~10 ka), according to the 14C ages by Roberts and Evans (2013) and to the OSL ages
provided in this work for the lacustrine deposits (Lac) (Fig. 16d).

5. Formation and permanence of the Seymareh Lake (~10-6.6 ka), according to the [14]C ages by Roberts and Evans, 2013 and to the OSL ages provided in this study for the lacustrine deposits (Lac) (Fig. 16e). The progressive infilling of the lake reservoir reduced progressively the infiltration section on the upstream side of the landslide dam. It cannot be excluded the presence of a minor emissary on the downstream side of the landslide debris.

6. Overflow of the lake and cut of the natural dam with formation of a first strath terrace (6.59±0.49 ka), followed by a second strath terrace and a flood plain during the emptying of the lake, which upstream is associated to the sedimentation of a fluvio-lacustrine sequence at the top of the lacustrine sediments (Fig. 16f).

7. Complete emptying of the lake and generation of the suite of fill terraces entrenched in the deposits of Seymareh Lake (4.5 ka. - Present) (Fig. 16g).

**7 Conclusion**

In a multi-modelling approach to the study of MRC processes affecting slopes at a large space-time scale (Martino et al., 2017), the analysis of geomorphic markers allowed us to constrain the landscape evolution of the Seymareh river valley in north-western Zagros Mts., before and after the failure of the largest landslide ever recorded on emerged Earth surface, provided a time scan of the main stages of valley evolution. The revised geological setting as well as the identification and
dating of different suites of lacustrine, alluvial and strath terraces, highlighted the predisposing factors that led to the gravitational instability and the geomorphic effects on the valley system. OSL ages obtained for eight samples, along with



the [14]C age provided by Roberts and Evans (2013), allowed us to provide significant time constrain to the valley evolution, which can be summarized in the following six points.

1. The oldest preserved geomorphic markers in the Seymareh river valley are represented by relict conglomerates upstream of the landslide, which testify to the early (Pliocene?) position of a Paleo-Seymareh River flowing into a synclinal valley close to the north-eastern flank of the Kabir-kuh fold.

2. The drainage re-organization associated to the growth of the north-western Zagros folds is testified by a major "slope-break knickpoint" along the Seymareh longitudinal profile, whose retreat during the Middle-Late Pleistocene is marked by the development of a suite of four alluvial terraces. Such a knickpoint retreat reached the portion of the Kabir-kuh fold that ~10 ka was affected by the Seymareh landslide. The collapse was therefore prepared by MRC processes acting over a time window in the order of $10^2$ ky;

3. Among the predisposing factors, the above mentioned erosional wave was accompanied by the deep incision of the gully network developed on the slope, which, along with the Seymareh river erosion, released kinematically the rock mass. The geological structural setting of the fold sector affected by the landslide, with respect to the adjacent ones, represents a further predisposing factor.

4. The geomorphic response to the landslide dam consisted in the formation of three lakes, among which the Seymareh lake persisted for ~3500 years before the beginning of the emptying phase started ~6.6 ka due to lake overflow. A sedimentation rate of 10 mm y[-1] was estimated for the lacustrine deposits, which increased up to 17 mm y[-1] during the early stage of lake emptying due to the increased sediment yield from the lake tributaries.

5. Since ~4.5 ka a suite of four alluvial terraces upstream of the landslide testify to the alternating erosion/deposition phases of the re-established Seymareh River.

6. An incision rate of 1.8 cm y[-1] was estimated since the beginning of the landslide cut by the Seymareh River and such a strong erosion started propagating up to the landslide source area where badlands catchments developed, eroding a volume of ~0.1 km[3] in the marly Pabdeh-Gurpi Fm..Such an evidence provided new insights on the geometry of the sliding surface, which likely did not involve the Pabdeh-Gurpi Fm., as inferred by Roberts and Evans (2013).

The here obtained results provide main constraints to the slope-to-valley floor evolution that can be very useful for numerical modeling of time-dependent stress-strain evolution of the Seymareh landslide slope. Such a numerical modeling could be the focus of future researches with the aim of calibrating the rock mass viscosity and verifying the possible earthquake trigger of the Seymareh landslide as an ultimate scenario of an ongoing mass rock creep process which reasonably was affecting the slope since $10^2$ ka.

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



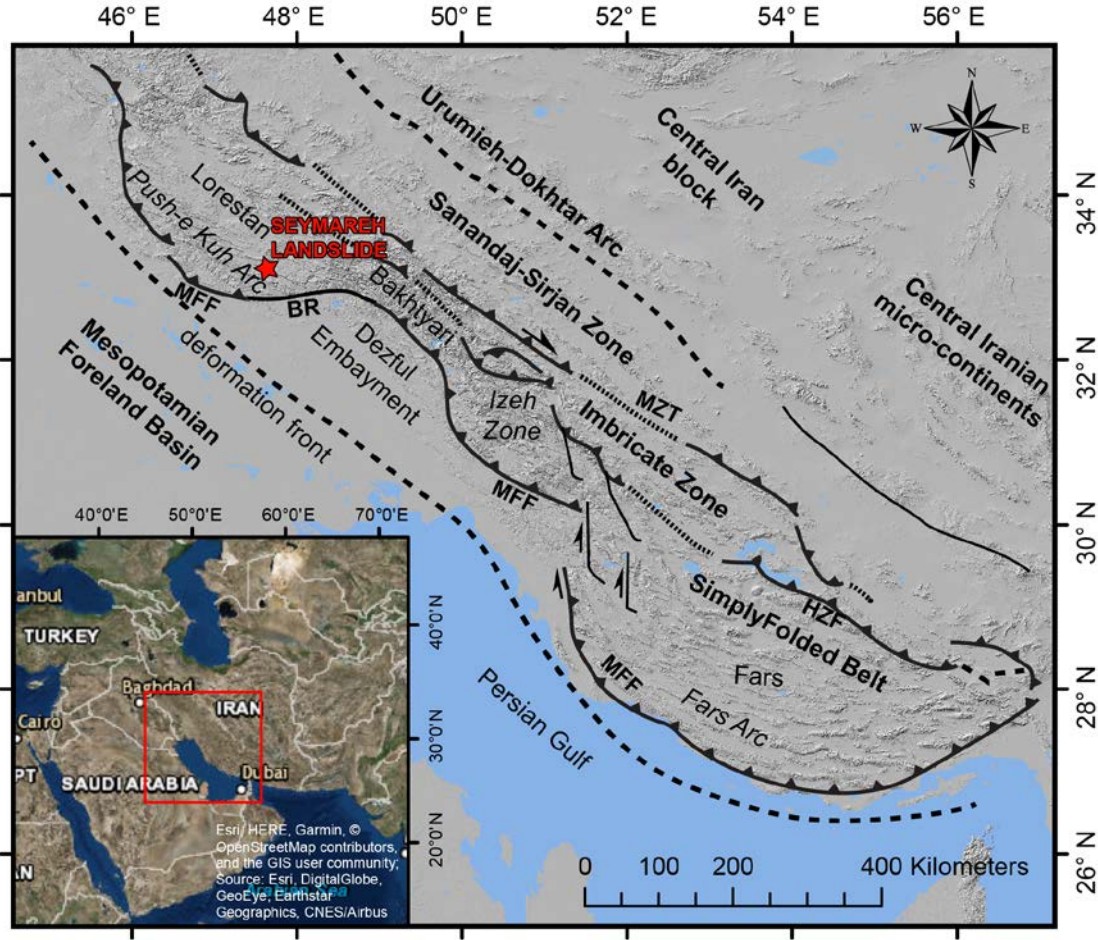

**Figure 1: Simplified structural map of the Zagros mountain range with location of the Seymareh landslide. MZT, Main Zagros Thrust; HZF, High Zagros Fault; MFF, Mountain Front Fault; BR, Bala Rud fault zone (modified from Casciello et al., 2009).**

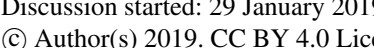



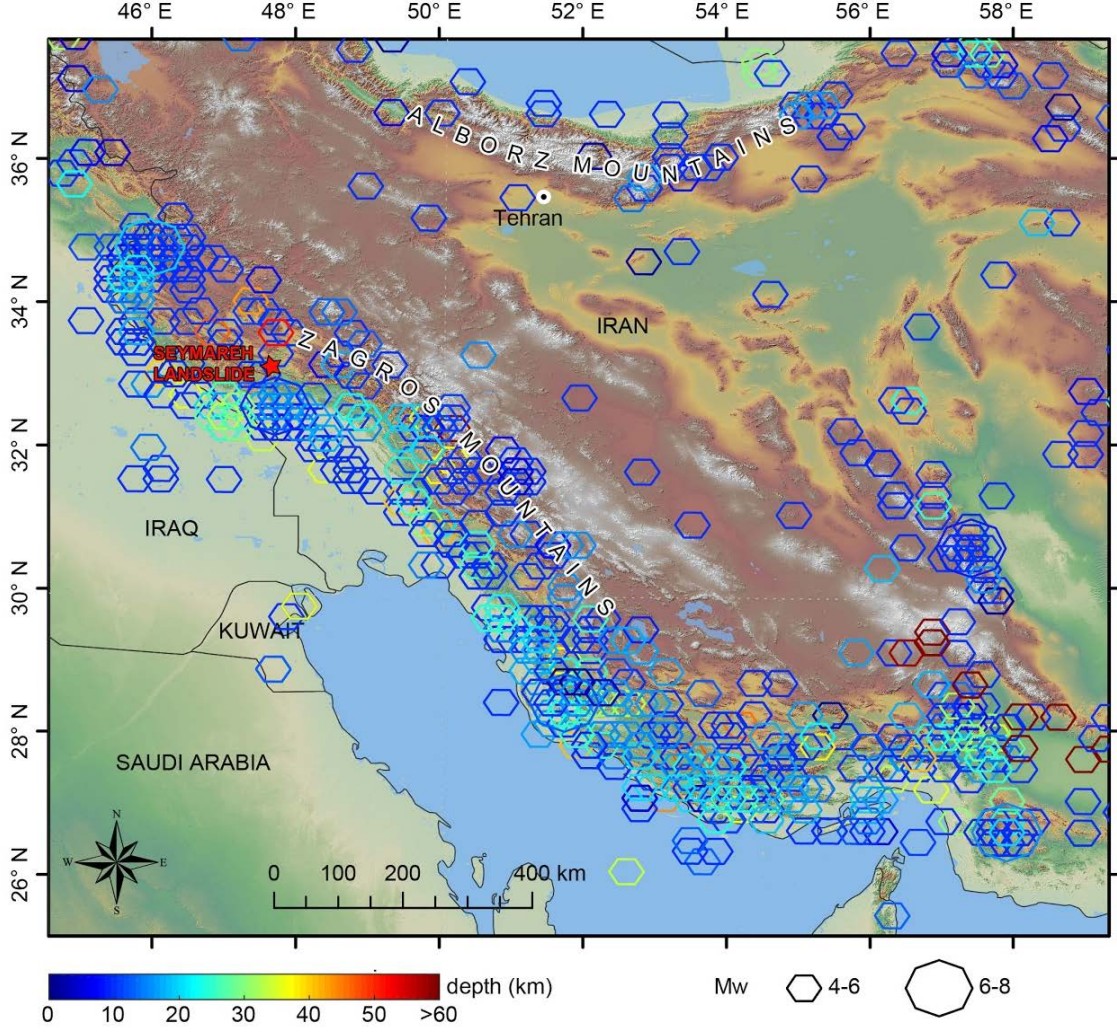

**Figure. 2: Magnitude and depth of the recent earthquakes recorded in the Zagros Mountains (source: IRIS Earthquake Browser, https://www.iris.edu/hq/inclass/software-web-app/iris_earthquake_browser).**



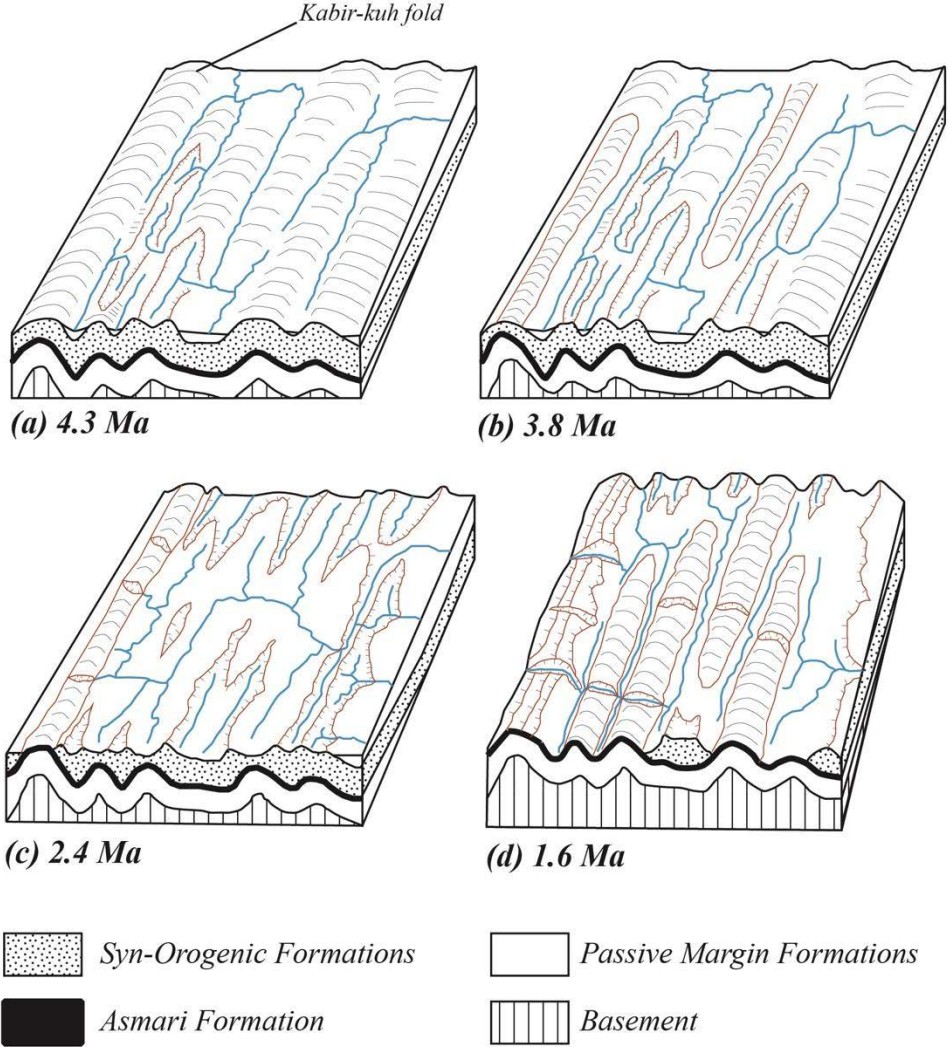

**Figure 3: Evolution of the drainage network in the Zagros chain sector of the Kabir-kuh fold, according to the Oberlander's model and with the timing provided in the landscape evolution model by Tucker and Slingerland (1996). See the text for explanation of the four steps. (modified from Oberlander, 1985).**

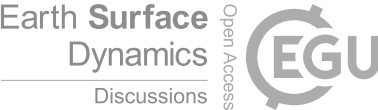



**Figure 4: Overview and focus (in the red box) of the Seymareh rock avalanche, Zagros Fold–thrust Belt, NW Iran. (modified from Google Earth®).**



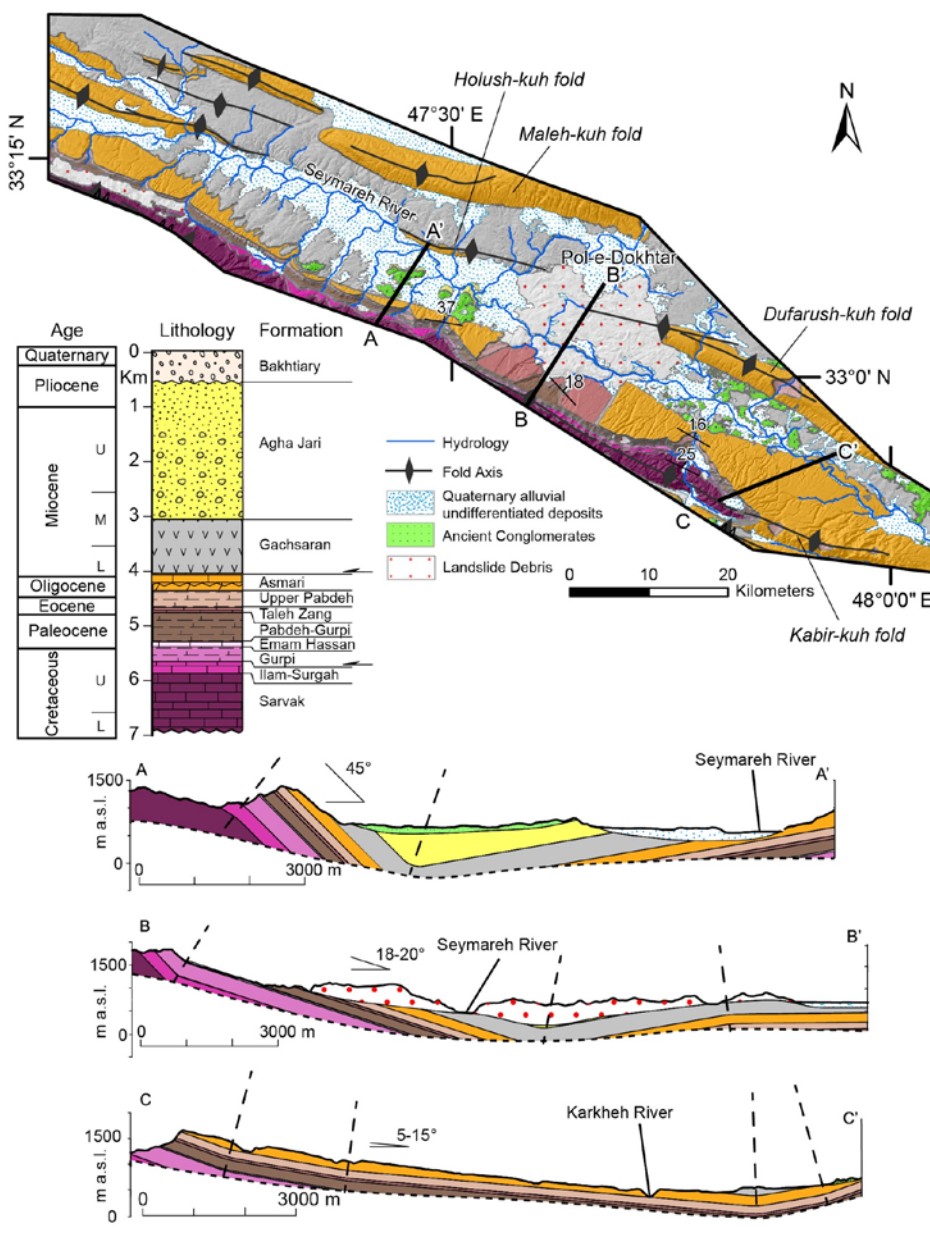

**Figure 5: Geological Map and related cross sections of the study area according to the revised stratigraphic column for the Seymareh River valley.**





**Figure 6: The suite of four orders of alluvial terraces entrenched in the lacustrine deposits (Lac) of Seymareh Lake upstream of the landslide, in the areas where Harrison and Falcon (1938), Roberts and Evans (2013) and Shoaei (2014) hypothesized the natural damming lake could be extended. a) Overall view of the suite of terraces; b) example of fluvial terrace deposit; c) example of lacustrine deposit.**



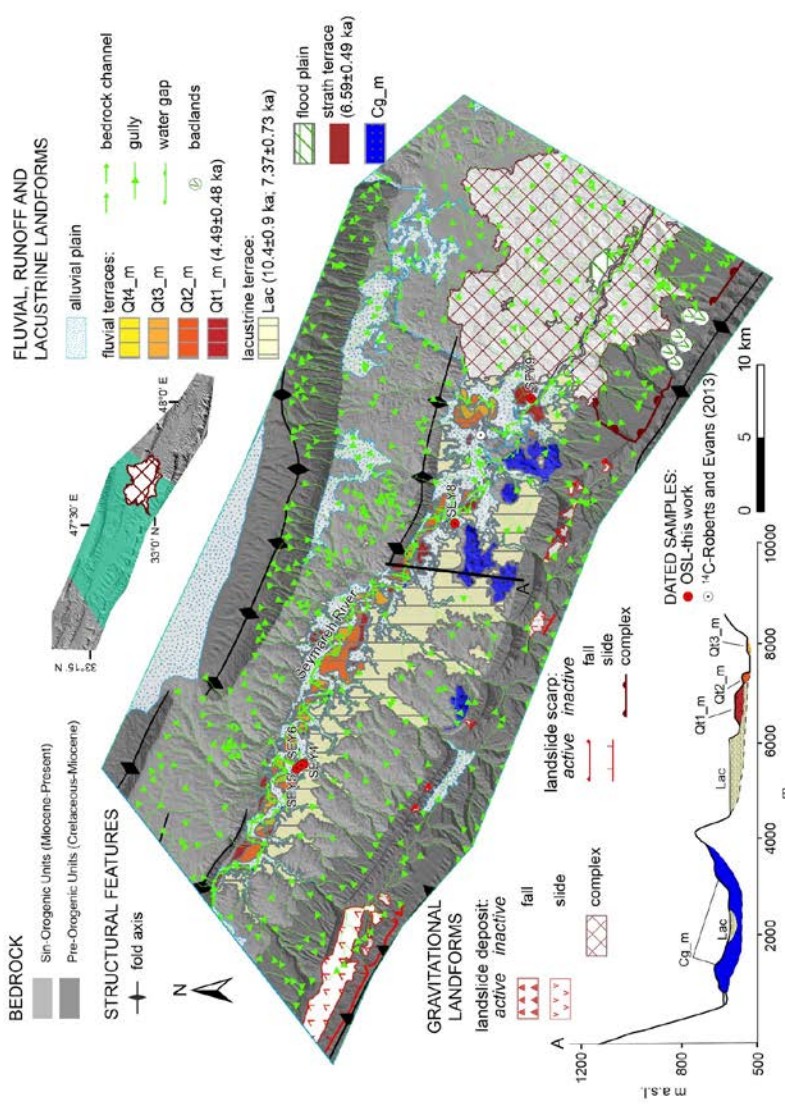

**Figure 7: Map of the Seymareh Lake alluvial and lacustrine terrace suite and of the most significant landforms for the valley slopes evolution.**





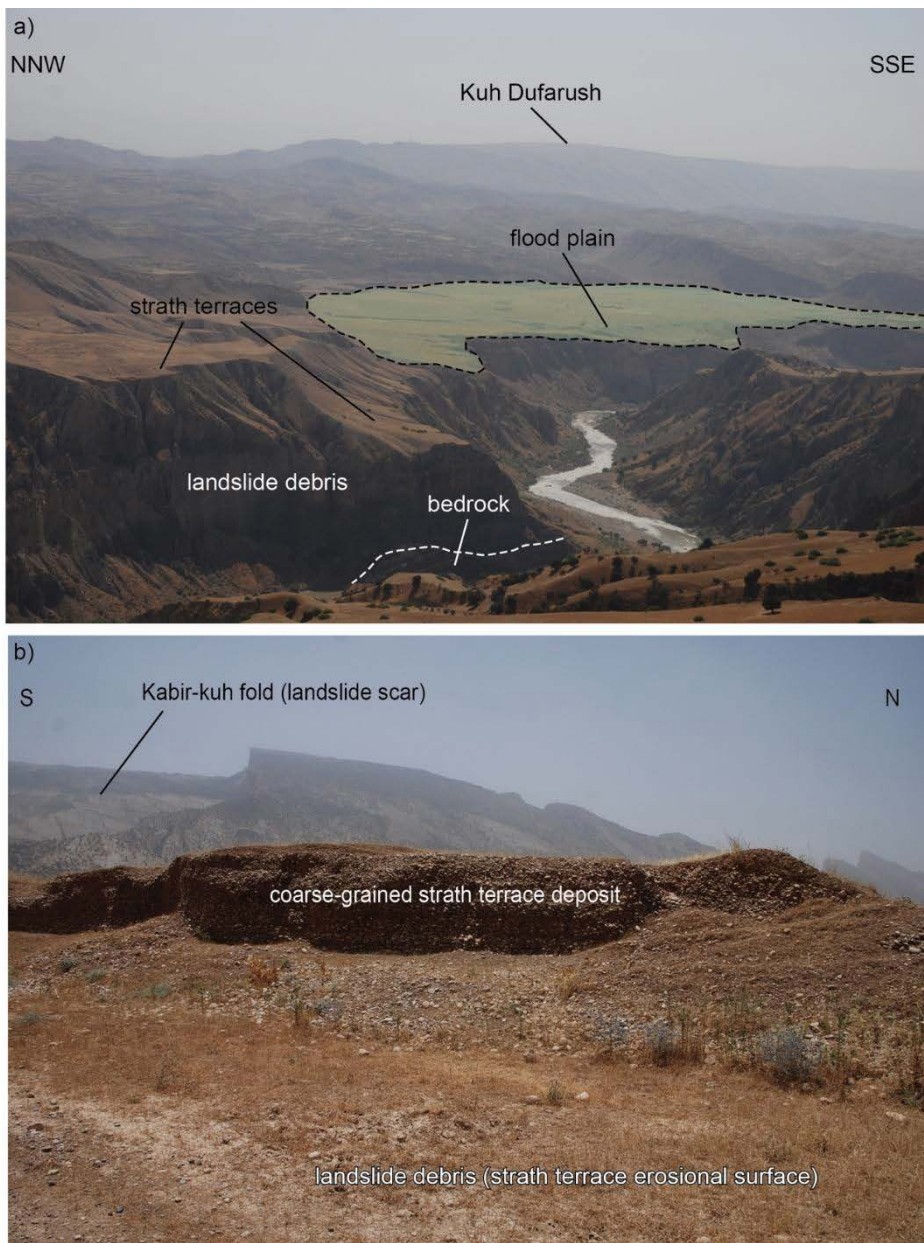

**Figure 8: a) Strath terraces and flood plain developed over the landslide debris, which are important markers of the evolution of the natural dam. In fact, they testify to the moment of the overcoming of the damming lake and the overflooding of the river onto the landslide debris, respectively. Both the landforms are delimited by steep scarps caused by the fast engraving due to the Seymareh River that cut the entire landslide thickness and the underlying bedrock; b) detail of the strath terrace deposit sampled for OSL dating.**



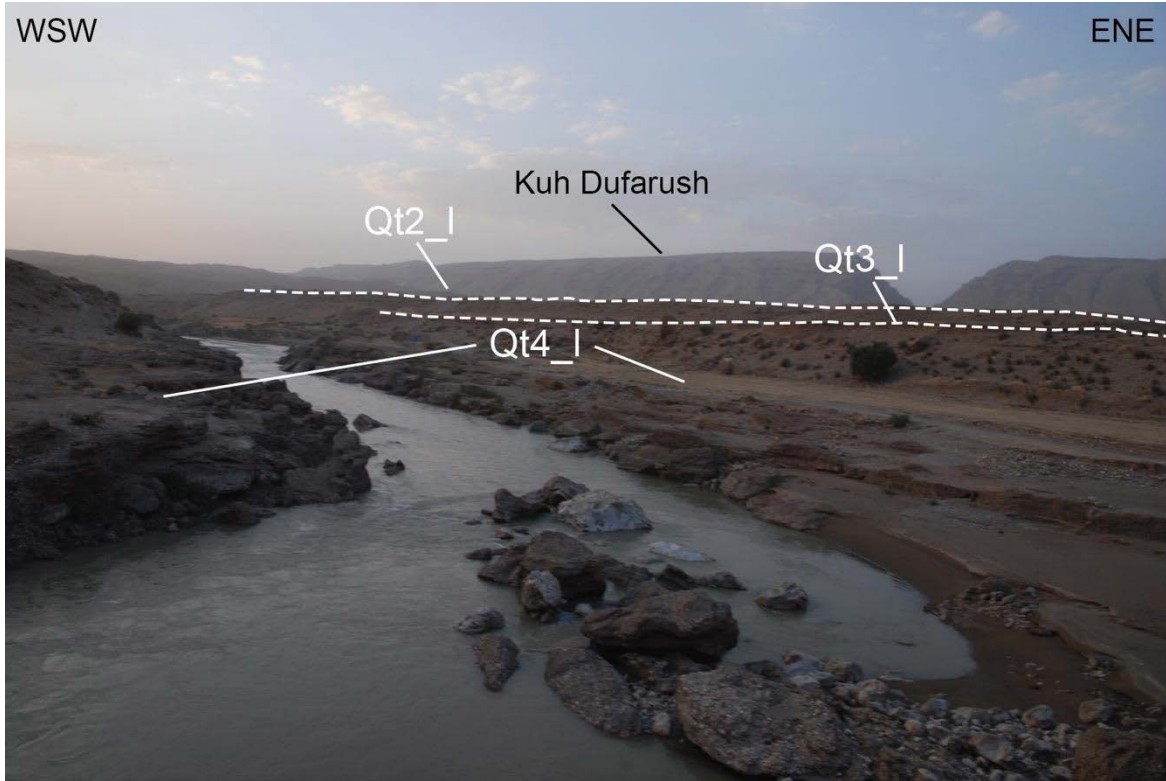

**Figure 9: The suite of fluvial terraces downstream of the Seymareh landslide. The Qt1 level is poorly preserved and not visible in this photo.**



**Figure 10: Map of the alluvial terrace suite downstream of the landslide and of the most significant landforms for the valley slopes evolution.**





**Figure 11: Projection of the pre-failure geomorphic markers (upstream and downstream conglomerates; downstream fluvial terrace suite) along the longitudinal profile of the Seymareh River and the age constraints resulting from the OSL dating.**



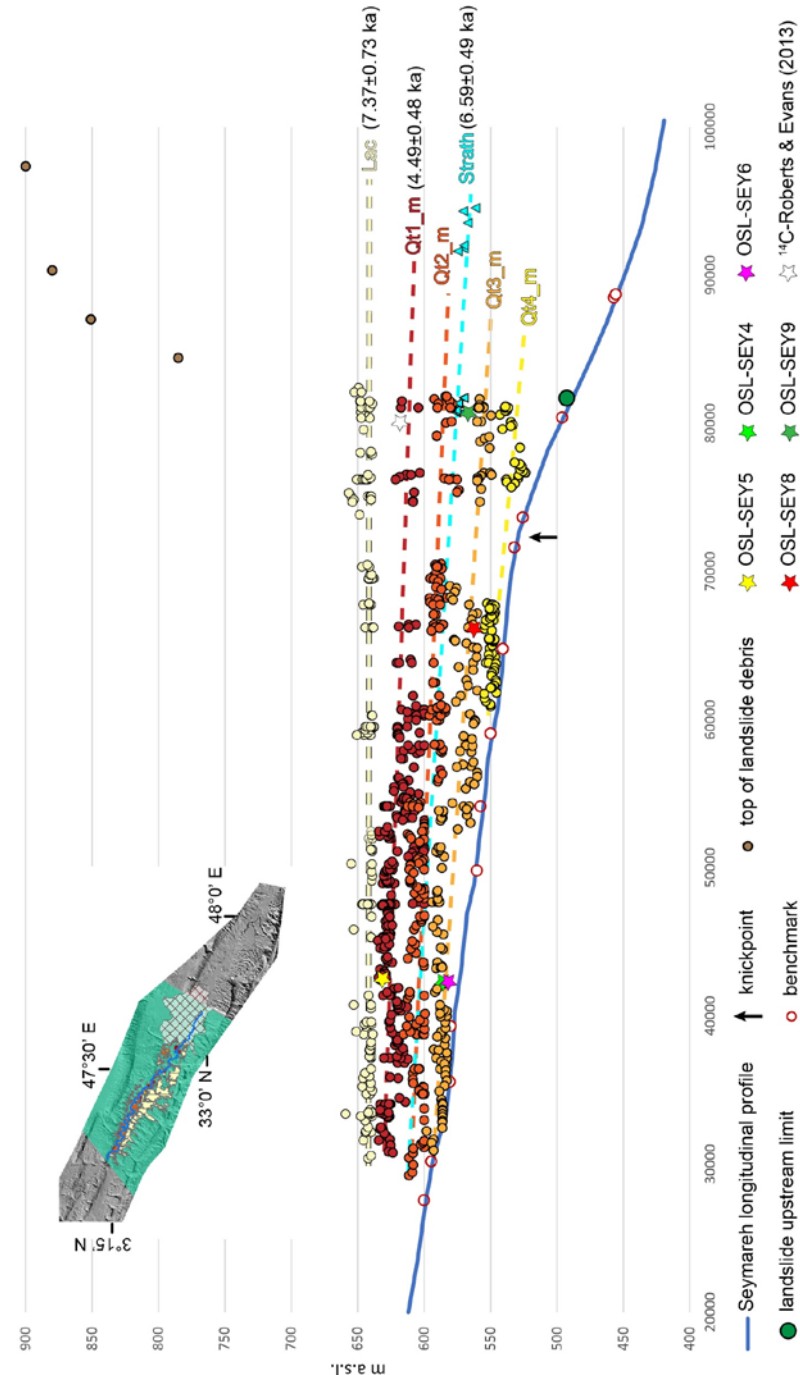

**Figure 12: Projection along the longitudinal profile of the Seymareh River of effect markers (fluvial terraces and lacustrine deposits relative to the Seymareh Lake suite) and the age constraints resulting from the OSL dating.**





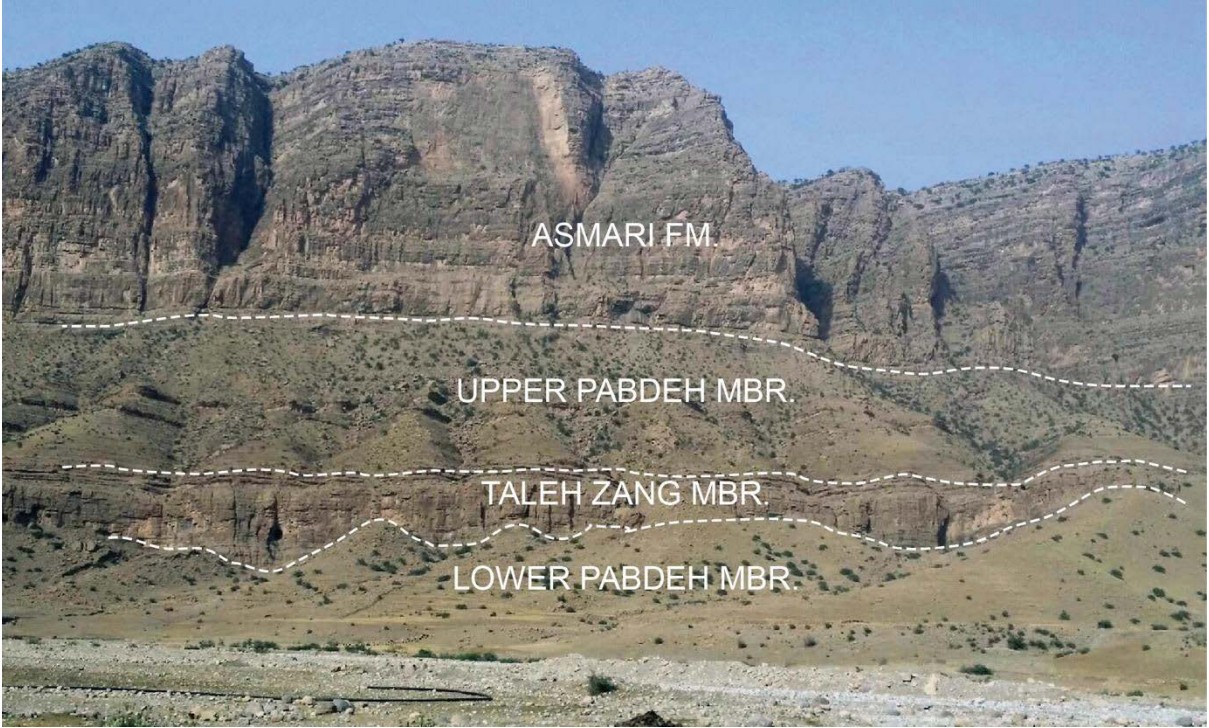

**Figure 13: View of the outcropping geological succession responsible for the development of MRC process. The different response to erosion likely corresponds to a rheological contrast.**







**Figure 14: Evidence of MRC processes. a) Front view of the scar area with the location of sites where evidence of buckling has been recognized; b) ductile buckling deformation of the Upper Pabdeh Member; c) brittle buckling deformation of the Taleh Zang Member along the scar area.**

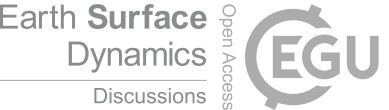



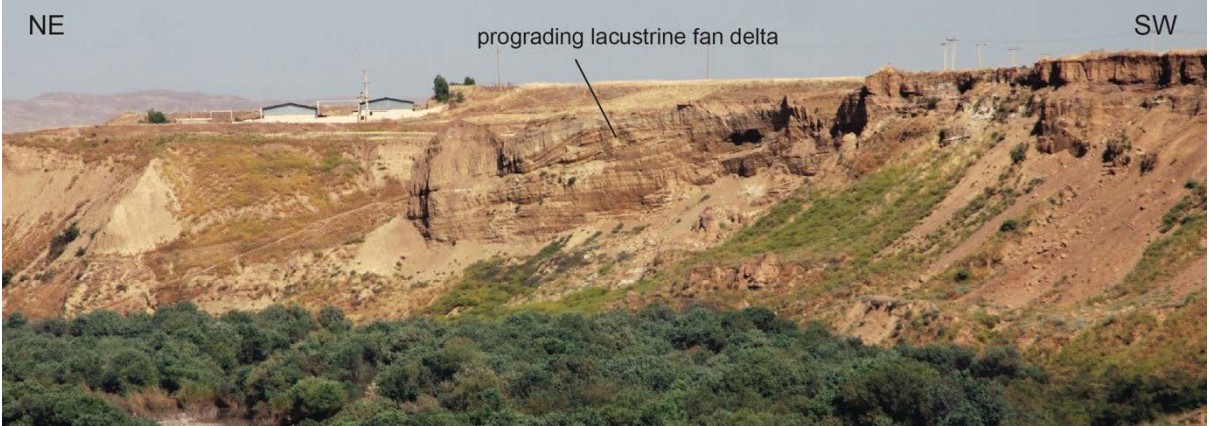

**Figure 15: Evidence of a prograding lacustrine fan delta formed by one of the right tributaries of the Seymareh lake during the emptying phase of the lake.**





**Figure 16: Evolutionary model of the Seymareh river valley. See text for explanation. Traces and legend of geological cross-sections are reported in Fig. 5.**



**Table 1: OSL ages obtained for the geomorphic markers recognized in the Seymareh river valley. Detailed information and photos of sampling sites are available in the supplementary material.**

| SAMPLE | DESCRITPION | COORDINATES | ELEVATION (m a.s.l.) | OSL AGE (ka) | ERROR (ka) |
|---|---|---|---|---|---|
| SEY4 | lacustrine deposit | 33° 13.197'N 47° 18.382'E | 590 | 7.37 | ±0.73 |
| SEY5 | alluvial terrace deposit (Qt1_m) | 33° 13.437'N 47° 18.219'E | 607 | 4.49 | ±0.48 |
| SEY6 | alluvial deposit beneath the lacustrine deposit | 33° 13.291'N 47° 18.358'E | 580 | 17.9 | ±1.50 |
| SEY8 | lacustrine deposit at the base of Qt2_m | 33° 7.402'N 47° 28.795'E | 560 | 10.4 | ±0.90 |
| SEY9 | deposit of a strath terrace on landslide debris | 33° 4.462'N 47° 34.197'E | 570 | 6.59 | ±0.49 |
| SEY3 | alluvial terrace deposit (Qt2_l) | 32° 59.591'N 47° 46.144'E | 485 | ≥373* | ±34 |
| SEY10 | alluvial terrace deposit (Qt3_l) | 32° 59.335'N 47° 46.071'E | 436 | ≥312* | ±45 |
| SEY11 | alluvial terrace deposit (Qt4_l) | 32° 59.265'N 47° 45.869'E | 400 | 60 | ±5 |

