# Peer review of "Reconstruction of river valley evolution before and after the emplacement of the giant Seymareh rock avalanche (Zagros Mts., Iran)"

_Earth Surface Dynamics, 2018_

## Referee Comment (RC1) · Nicholas Roberts (Referee) · 17 Mar 2019

The study investigates valley-bottom evolution of the Seymareh River valley bracketing the gigantic, prehistoric Seymareh landslide. If contributes new details on drainage variability, particularly changes in response to the formation and later erosion of the landslide dam, supported by new geochronologic controls. The manuscript does not, however, provide new insight on causes of the landslide or its numerical modeling, despite these topics appearing repeatedly in the abstract, introduction and conclusions.

[Figure]

This has the potential to be an interesting contribution to understanding of 1) the Seymareh landslide and 2) landscape evolution of the simply folded zone of the Zagros fold-thrust belt. These are very welcome additions as both subjects have received less attention than they deserve. The authors' focus on long-term valley-bottom geomorphic change is particularly interesting as such an approach has received minimal attention in past examinations of this landslide.

The main contribution of the paper – valley-bottom geomorphic evolution – fits with the journal's focus and is as far as I know unique to any study conducted in Persia.

The manuscript unfortunately suffers from numerous deficiencies that prevent me from being able to recommend it for acceptance in its current state. It could be reconsidered following very substantial reworking, additions, and improvements. The greatest issues include:

i. Overall imbalance of content, with over half of the manuscript comprising background material;

ii. Unnecessary details and focus on some topics (numerical modeling, pre-failure creep, seismicity and hazard/risk) that take up large parts of the abstract and/or introduction, but that are not part of the current contribution and are not directly relevant to its conclusions;

iii. Incomplete review of previous work on the Seymhareh landslide and geology of Kabir Kuh, including misattribution of several findings and interpretations;

iv. Insufficient methodologic descriptions that prevent the new work from being properly evaluated or replicated;

v. Confusing organization that includes: a section purported to present new material (section 4) comprising largely a repeat of what was already known; a results section (section 5.2) containing a mix of observations/interpretations and results; and a discussion (section 6) presenting apparently new observations/interpretations.

**ESurfD**
vi. Writing that is commonly overly wordy, confusing, or incorrect.

The major issues, including various aspects of the six points above, are listed below. These are followed by minor issues, which are predominately examples of language and writing issues. The list of minor issues is not comprehensive, but provides an idea of the types of problems that should be addressed throughout the manuscript.

MAJOR ISSUES

1. The writing is very wordy and convoluted. Many sentences are unnecessarily long and complex, making them hard to follow. Grammatical and language errors are abundant. The scale of these problems make substantial rewriting necessary, and thus in my view is a major issue. Many examples of such instances are provided in the list of minor issues below, but this is not an exhaustive list.

2. The title does not accurately convey the main thrust of the paper. It suggests that the paper provides new understanding of the causes of the landslide, which it does not. This paper is about valley-bottom geomorphic evolution before and after a gigantic landslide.

3. The abstract does not summarize the present study well, and instead mentions all sorts of things that are not part of the authors' work; although some of these are peripherally related (hazard, seismic triggering, causes of the landslide) they do not constitute anything new as far as I can tell from the presentation of the rest of the paper. At the same time, the abstract lacks details about some of the major interpretations and results from the body of the paper. It needs to be rewritten and streamlined.

4. The summary of previous work on the Seymareh landslide is missing many key points and attributes some details to the wrong sources. Other details are not attributed at all. For instane:

a. Page 3, line 6: Some important and very relevant contributions of Roberts (2008) and Roberts and Evans (2013) are not mentioned. Those sources propose a detailed
model of how the geologic and tectonic evolution of Kabir Kuh predisposed the slope to such large-scale failure, including formation of structural/kinematic and rheological control. As far as I can see, this contribution is not recognized in the current paper, despite it being directly related to the authors' claimed contribution of improved understanding of factors predisposing the slope to gigantic failure.

b. Page 3, line 9: It is not sufficiently clear that the age estimate of 9800 radiocarbon years is based on the interpretation of three separate radiocarbon ages. This 9800 a BP age is taken from Roberts and Evans (2013) and must be cited accordingly. The ages provided by other sources - at least those from Griffiths et al. (2001), which were not influence by the 'hard-water effect' – should also be noted so that the reader does not have to refer back to Roberts and Evans (2013).

c. Page 3, line 6: Yamani et al. (2012) provide no new details on emplacement mechanisms of the landslides (at least not from the details in the English language extended abstract of their paper writing in Farsi). If there are some details missing from the Yamani et al.'s English text that they present manuscript refers to, it would be very helpful to provide translated quotes in the supplemental material. Otherwise, mention of new details on landslide emplacement attributed to that source needs to be removed. Yamani et al.'s (2012) main contribution comprises some general details on the evolution of lake drainage.

5. The paper includes a lot of largely unimportant, or at least overly specific, background details. Many of the details about tectonic features and some of the details about seismicity in section 2 are well beyond what is necessary to provide relevant background to the reader. These extraneous details could appear in the supplement to provide further context for the interested reader, but they take up too much of the main paper. Given that modeling is not part of the present paper, much of the background provided about modeling is irrelevant.

6. If the mechanisms and behaviour of large bedrock landslides are to be discussed,

some recent reviews pertaining to this topic should be cited (principally Brideau and Roberts [2015], which includes this event as a case study, and Hermanns and Longva [2012]). Those sources provide numerous additional references on progressive failure that should also be considered if the authors can make a suitable case for discussing this topic.

7. Much of the background information lacks proper referencing, including tectonic setting (section 2), seismicity (section 2), and geologic setting (section 3). For example, the paragraph starting on page 4, line 27 presents many details that are clearly not part of the current study, but only provides references in two places. Furthermore, this generalization of sources does not allow the reader to sort out what details have come from what sources.

The hypothesis cited in the final paragraph of section 2 is based on much more than just the frequency of strong earthquakes in the region, as this submission suggests. Progressive steepening of the slope at a very slow rate relative to the modern recurrence rate of nearby strongly felt earthquakes is a crucial consideration as it makes failure initiation in the absence of seismic loading hard to explain.

8. Thickening of the Pabdeh Formation (top of page 6) very well may have in influence on landscape development. However, the position of the northwest-trending deformation front also strongly inferences the westward change in landscapes. This was also recognized by Oberlander in his work, but appears to have been overlooked.

9. In contrast to what the authors claim, I see no evidence that this work has revised the stratigraphy of the study area to any substantial degree. The stratigraphy of the part of this Zagros fold-thrust belt is extensively reported in previous work (much of which as not been referenced here). The stratigraphy described here matches very closely (in sequence order, composition, and thickness) with the stratigraphy already reported in the literature. For instance, I see very little difference between the sequence in Fig. 5 of the present manuscript and the stratigraphy summarized from review of existing

literature by Roberts and Evans (2013, their Fig. 5).

Several important references on the sequence are missing and should be included. For the overall region this includes James and Whynd (1965) and Alavi (2004). Detailed mapping of Kabir Kub conducted by Iran Oil Operating Companies (Setudehnia and Perry, 1967; Takin et al., 1970; Macleod, 1970) already covers much of what the authors would have covered in their 'new' mapping presented in section 4 and in Fig. 5.

The authors need to first clearly describe what has already been documented (in a section on geologic background). Only after that should they try to justify how their 'new' stratigraphy differs. From what I can tell, they contribute only some additional detail on the Pabdeh Formation, although at least some of this is similar to that reported in the Iran Oil Operating Companies maps. A much more convincing argument will need to be made if any new contribution to the area's stratigraphy/geology is to be claimed.

The details presented in section 4 nearly all belong in the background material. This further highlights an issue with the paper's layout and balance: the new contribution of the 15-page manuscript doesn't start until page 9, meaning that over half of the paper is background information.

10. The Gachsaran Formation has a high gypsum content, which dominates its geomechanic and geomorphic behaviour. The description provided here (Page 8, line 17) instead suggests that it comprises only more typical clastics. Given this manuscript's focus on valley-bottom evolution, this unit needs to be accurately characterized.

11. The details, or even relevance, of anticline flank dips (second paragraph of section 4) is unclear. Dips along the three sections probably need to be considered more carefully. Due the nature of the anticline, structural variation is to be expected. Dips will of course decrease to the southeast toward the nose of the anticline. What is possibly more interesting is how much steeper dips are to the northwest beyond the

mapping presented here. Furthermore, this is a complex box fold with a hinge near the upper part of the Seymareh detachment zone. Downslope variation flank dips will reflect this structure.

Due to the complexity of the fold, stratigraphic position will also affect dip. For example, Roberts and Evan (2013) noted steepening (see their Fig. 9H for example) of the Eman Hassan member in the upper, central part of the landslide scar. In contrast, the adjacent upper surface of the Asmari Formation on either side of the landslide is much less steep. Variations between other units are noted also in mapping by Iran Oil Operating Companies (summarized in part in Fig. 4B of Roberts and Evans). Such variation has important implications for dip-slope failure of the flank.

In light of these points, it would thus be very helpful (and far more informative) if the authors provided a detailed map of their structural measurements, and clarify how these may build upon those provided by past studies of the landslide and by mapping by Iran Oil Operating Companies and by Roberts and Evans (2013). Conversely, if these measurements are based only DEM profiles, that needs to be clearly stated. In any case, how well profiles through the eroded core of the anticline represent protections the Asmari limestone limbs beneath the valley floor needs to be evaluated given stratigraphic variation in dips relating to the complex nature of the flod.

Finally, profile C-C' is oblique to the true dip of the anticline flank and appears to thus under represent dip of the Amsari surface (based on Fig. 5, it appears that the apparent dip, not the true dip, is being represented).

12. Is this the reconstruction of the Asmari carapace (Page 8, line 29) immediately pre-failure or a reconstruction of the anticline prior to unroofing? Please clarify. In the latter case, 2100 m a.s.l. is a substantial underestimate and the structure of the box fold suggests that the Asmari carapace extended much higher.

13. The authors' point about the position of Seymareh River (Page 8 line 31) does not become clearly relevant later one in the paper. Depending on its significance for

the current study, which remains unclear, the authors will need to clarify whether this river position is related to migration of the channel within the Lake Seymareh lacustrine deposits (i.e. following valley damming and sediment in-filling of the Lake Seymareh basin) or is the result of some older physiographic control. The river has obviously migrated to the northwest in the last several hundred years as a Sasanian-era bridge over the old river position at Pul-i-Shikari is nowhere near the modern river (see observation and discussion by Harrison and Falcon, 1938).

14. I cannot understand what the final five lines of section 4 mean. I assumed the kinematic release suggested here is that of the Seymareh landslide. What is the 'connectivity' supposed to be? How are the flatirons envisioned to control sliding? Note that several previous studies suggest fluvial undercutting of the slope as the source of kinematic freedom at the slope toe. However, breakout across the upper units is also necessary for the failure to have occurred as the failure surface cuts stratigraphically upward in the downslope direction (Roberts and Evans, 2013); this feature is an important part of the kinematic release.

15. The methods section lacks sufficient detail. What are the source and scale of the air photos used? Were they interpreted quantitatively or only qualitatively? What specific imagery was used from Google Earth (there is of course a very wide range of imagery types and qualities available in that software)? What as the source of the map used? What inputs specifically were used the creating the DEM (current wording is unclear)? No methods are provided for the new geologic investigation that the authors claim to have conducted.

OSL can be a finicky technique. Many critical aspects of the sampling are not considered, particularly those necessary to rule out partial bleaching during sample collection and transport: was the slope cleared off first? was an opaque sample vessel used? how far was it inserted into the slope? OSL sampling methodologies vary quite a bit, so unless the approach exactly follows that of a pervious study, simply citing in past source here is insufficient.

16. The results section (section 5.2) is rather hard to follow and contains a mix of observations, interpretations and results. Furthermore, several observations/interpretations (conveyed in Figs. 14 and 15) are skipped over here and are referenced only in the discussion. The formatting of this section needs work. I see no benefit to using lists here (or anywhere in the paper); all text should be in paragraph format. Several parts of the lists are not even full sentences.

Issues with in the content include a lack of clarity over how the authors believe the dated sequences to relate to each other. For example, the wording at the start of the section (Page 10, line 9) seems to suggest that the lacustrine terrace pre-dates the landslide, but I cannot imagine how this is possible. Such as suggestion also conflicts with the post-landslide age reported for samples SEY4 and SEY5.

17. The discussion lacks any details about reliability of the OSL ages presented here. There are many possible error sources in this technique, but it is unclear if these were considered.

18. The discussion suggests that the current study provides some new insight on the geologic succession and its role in mass rock creek (MRC). I see no such contribution in the paper. The authors seeming claim in the abstract and introduction that their study somehow addresses pre-failure creep, but this topic is not investigated in any detail.

Pre-failure creep is hardly mentioned, and even then is based only a couple of field observations. The suggestion that features noted in the Pabdeh Formation (Page 13, line 2) indicate pre-failure creep is not sufficiently supported. It is also possible that these features are not a result of progressive failure of the slope. The plastic deformation shown in Fig. 14B could well be the result of pre-failure creep, but not enough detail is provided to evaluate this. Rock mass strength reduction and associated deformation is also to be expected as a result of fold formation (see Roberts and Evans, 2013 and references therein). The brittle deformation shown in Fig. 14C could well be the results of sliding during catastrophic failure, so I see no reason to use it to argue for pre-failure

creep.

19. The authors also bring up numerical modeling in the abstract, introduction and discussion, despite it not factoring into the study. Perhaps they mean to indicate that their results could be used to guide/inform numerical modeling, but their wording is unclear. Given that numerical modeling is not addressed in the study, the background on modeling in the introduction is too extensive and Figure 14 seems irrelevant.

20. From what I can tell, the authors provide no clear new evidence regarding pre-disposition of the slope to failure. They start to suggest two interesting aspects, but do not adequately address them. The first aspect is that of pre-failure creep (noted above). The second, which is more closely tied to their investigation, is the possibility of knickpoint migration along Seymareh River, which is inadequately communicated. The extensive evaluation geologic and geomorphic controls on the failure by previous studies (particularly Roberts and Evans, 2013) is hardly addressed, although authors of the present study claim that this is one of their main focuses.

21. Why are landslide kinematics described in the discussion (top of page 13) when they are not part of the new work presented here and when the detailed examination of this topics by previous workers is not mentioned in the background sections? For instance, the geomechanical strength contrast between the Asmari and Pabdeh formations is mentioned in the current paper, but has not been characterized. It is, however, approximated in Roberts and Evans (2013).

What evidence is there for kinematic freedom provided by gullies along the flank of Kabir Kuh (Page 13, line 13)? The lateral margins of the main landslide are nearly vertical features following a major joint set in the Asmari Formation (characterized in Roberts and Evans, 2013 and references therein, but not mentioned here). There is no evidence I can see for these being related to fluvial processes. Roberts and Evans (2013) propose that these features are instead inherited from the tectonic history of the Zagros' simply folded zone.

22. What is the basis for the suggestion that failure was preceded by an 'elapsing time' on the order of 100 ka (Page 13, line 17)? It's not even clear what this period is meant to represent. Is this a period of pre-failure creep? The period between the knickpoint passing the toe and the slope failure? This is very hard to follow.

23. The conclusion has several issues. It seems to include material – some kind of modeling related to the Seymareh River valley – that is not only not part of the present study, but as far as I can tell is not included in any other published research. I do not see how this fits in, other than also being mentioned in the abstract and introduction.

The list summarizing landscape evolution of this part of the Seymareh River valley seems very similar to the list presented on the previous page in the in the discussion, and is thus quite redundant.

The conclusion ends with a sentence about seismic triggering, which is hardly mentioned in the paper other than stating that pervious workers have suggested it (and even then fails to adequately explain what has been done before). I don't see how the new work done here will contribute to evaluation of a seismic trigger. This text seems to be irrelevant to the conclusions of the paper.

24. The figures are too numerous. Several can be combined (particularly the photos) and others seem to have limited relevance. Fig. 2 is probably more appropriate in the supplement given the lake of relevance of seismicity to the current study. Figs. 1 and 3 could probably be combined, especially if some of the extraneous detail in Fig. 1 is removed. Figures 13 and 14 are irrelevant to the main focus of the paper. Fig. 13 does not add anything to the paper. Figure 14 relates to the suggestion of pre-failure creep of the slope, which contrary to what the authors state in the introduction of the paper, is not a major component of the present study. Figure 15 seems to be an afterthought, although it has far more relevance to the paper's focus on valley-bottom geomorphology that either of the preceding figures.

The list below provides references for several works that the authors should consult

and that are missing from the current paper:

Alavi, 2004. Regional stratigraphy of the Zagros fold–thrust belt of Iran and its proforeland evolution. American Journal of Science, 304, 1–20.

Brideau and Roberts, 2015. Mass movements in bedrock,in: Landslides Hazards, Risks and Disasters, [Davies and Shroder, eds.]. Academic Press, Amsterdam,Netherlands, 43–90.

Griffiths et al. 2001. Environmental change in southwestern Iran: the Holocene ostracod fauna of Lake Mirabad. Holocene, 11, 757–764.

Hermanns and Longva, 2012. Rapid rock-slope failures, in: Landslides: Types, Mechanisms and Modeling [Clague and Stead, eds.]. Cambridge Univ. Press, Cambridge, UK, 59–70.

James and Wynd, 1965. Stratigraphic nomenclature of Iranian Oil Consortium Agreement Area. AAPG Bulletin, 49, 2182–2245.

Macleod, 1970. Kabir Kuh, 1:100000 Geological Map. Iran Oil Operating Companies, Geological Exploration Division, Tehran.

Setudehnia and Perry, 1967. Dal Parri. 1:100000 Geological Map. Iran Oil Operating Companies, Geological Exploration Division, Tehran.

Takin, M., Akbari, Y. & Macleod, J.H. 1970. Pul-E Dukhtar. 1:100000 Geological Map. Iran Oil Operating Companies, Geological Exploration Division, Tehran.

MINOR ISSUES

1. Page 1, line 10: The anticline if variously referred to as 'the Kabir-kuh fold', 'the Kabir-kuh Fold', 'Kabir-kuh fold', and 'the Kabir-kuh'. Kabir-kuh is a proper physiographic feature whereas the fold feature is not officially recognized as a name. Thus, the only proper version of the naming used here are 'Kabir-kuh' and 'the Kabir-kuh fold'.

2. Page 1, line 15: Proper capitalization and use of 'the' are: 'the Seymareh River valley' and 'Seymareh River'. This needs to be corrected throughout the manuscript.

3. Page 2, line 28: What is meant by 'different evolutionary stages'?

4. Page 2, line 28: 'allows to construct' is improper language.

5. Page 2, line 28: What are 'interesting valley sections'?

6. Page 2, line 33: The location description is incomplete. The Seymareh River valley straddels the border between Lorestan and Ilam provinces, respectively to the east and west of the river. The landslide initiated in what is now Ilam province, but most of the debris lies in Lorestan.

7. Page 2, line 34: What is meant by 'evolutionary scenarios'? 8. 9. Page 3, line 1: I am not familiar with the region 'External Zagros Mountains'. It this the simply-folded zone/belt?

10. Page 3, line 13: 'Seimareh' should be 'Seymareh'. Although various spellings have been used over the years, 'Seymareh' seems to be the currently recognized version. In any case, spelling should be consistent throughout the manuscript. This corps up in a few other places.

11. Page 3, line 15: It is far more informative to state here what the study achieved, rather than what it intended to achieve. Also 'aims at better understanding' is improper English.

12. Page 3, line 16: Here, in the abstract and again later on risk (or risk mitigation) is thrown in. However, this topic is not explored. Practically, the only mitigation would be complete evacuation (either temporality based on some kind of waring system or permanently) of an area that could experience landslide of this magnitude and stabilization or localized avoidance are impossible. Furthermore, the very low probability of a landslide of this magnitude means that it risk is potentially rather low.

[Figure]

13. Page 3, line 25: What is 'e.g.' used? Also, this is hardly the more relevant source for this statement given that several studies have investigated the landslide.

14. Page 3, line 27: Units should be m a.s.l. as this is an elevation.

15. Page 4, line 1: 'The Zagros' is not the proper way to refer to the range. This statement also requires references to back it up.

16. Page 4, line 15: 'landslide' not 'Landslide'.

17. Page 4, line 15: Given that this is a new paragraph, reference back to the previous content (using 'latter') is a confusing.

18. Page 4, line 22: Parentheses missing around publication year.

19. Page 4, line 22: What is meant by the 'onset of the [sic] deformation'? Is this supposed to be propagation of the deformation front?

20. Page 5, line 13: I've never come across 'Delful Zagros' as a term. Are the authors certain that this is a properly recognized physiographic region?

21. Page 5, line 14: Missing year of source.

22. Page 5, line 21: The mobile and competent units have not yet been introduced, and are part of the geology not the geomorphology (as the section title would suggest). These have not yet been introduced. The geology should be briefly summarized before the geomorphology, especially given the apparent influence of the former on the latter. A few of the units are mentioned in the following lines, but the geology is of course much more complex than that.

23. Page 5, line 22: This fold is in Ilam province, not Lorestan. The boarder between them in this area follows Seymareh River.

24. Page 6, line 28: Should be 'the Asmari Formation'. Spell formation out throughout the text of the paper; do not abbreviate to 'Fm.' (as on line 31).

25. Page 6, line 5: This is general physiographic background that should appear much earlier on in the paper.

26. Page 6, line 6: Identify the lakes here and how they were formed (i.e. which rivers were blocked). Lake Balmak is not named until the discussion and is hard to place in the figures. It would also be helpful to very briefly note that much of the previous literature calls this lake Chah Javal.

27. Page 6, line 9: Regarding '. . .formed in response to a sequence of landslide', clarify whether this is multiple separate landslides or all related to the Seymareh landslide.

28. Page 6, line 9: Consider simplifying to 'the landslide dams' so that the reader does not mistake your meaning to be multiple landslide dams of Seymareh Lake (instead of multiple lakes dammed by the Seymareh landslide).

29. Page 7, line 12: Combining 'none' and 'neither' forms a double-negative. Also, 'study' should be 'studies'.

30. Page 7, line 13: Specify 'fluvial' geomorphic markers (and remove 'the').

31. Page 7, line 17: 'landslide' not 'Landslide'.

32. Page 7, line 18: Presumably 'refer to' means 'date to'?

33. Page 7, line 19: Geologic ages should be late/early, not upper/lower. The latter pertain the stratigraphy, not ages.

34. Page 7, line 19: The ages should be 'Late Cretaceous' (an officially recognized age) and 'early Miocene' (not an officially recognised age).

35. Page 8, line 21: Unless I've missed a break, the paragraph ending on this line is massive and needs to be broken up.

36. Page 8, line 29: Do not abbreviate 'Formation'.

37. Page 8, line 34: The river name should be 'paleo-Seymareh river' as this is not an

officially recognized name.

38. Page 9, line 12: 'literature' is insufficient. What were the sources?

39. Page 9, line 18: Unclear what is meant by the 'data where [sic] acquired'.

40. Page 9, line 20: Write 'minutes' out in full.

41. Page 10, line 4: Which components of the Hydrology toolbox were used?

42. Page 10, line 17: Should be 'the Seymareh River gorge'.

43. Page 11, line 10: Why is this knickpoint '...the most interesting...'? The authors seem to be implying that this may related to instability within the flank of Kabir Kuh, but the reader can only guess.

44. Page 13, line 20: Lake Balmak is mentioned here for the first time. Why?

45. Page 13, line 24: What about the drainage was progressive? The actual drainage is now well characterized here.

46. Page 13, line 28: This duration for the lake needs to be compared with estimates provided in other sources.

47. Page 13, line 30: This is the only place this figure is cited. The figure does not alone indicate what the authors suggest. Has some further work been done on this stratigraphic section that I missed?

48. Page 14, line 3: What is meant by 'time scan'?

49. Page 14, line 5: I again see no benefit to a list instead of writing out the description in proper paragraph format.

50. Text in some figures, particularly the labels (and markers) for sample locations in Figs. 7 and 10, is too small and thus hardly legible.

**ESurfD**

Interactive
comment

---

## Referee Comment (RC2) · David Petley (Referee) · 1 Apr 2019

The Seymareh landslide is zone of the largest terrestrial rock slpe failures identified to date. Whilst there have been some interesting studies of the landslide, it remains somewhat under-investigated. Tom this extent the premise of this paper is good. The title of the paper is about predisposition and about geomorphic response, both of which are interesting topics in the context of this landslide.

However, the paper does not meet the standards required in its current form. I have

a number of fundamental issues: 1. The manuscript is very poorly organised. It is hard to understand what is new, what is a reinterpretation, and what is background information. I found the paper difficult to read and to follow, and at the end I am not sure i really managed to work out what was new. 2. There is a huge amount of background information. Much of this seems to be irrelevant or tangential. In some cases it misrepresents the literature (e.g to say "which is mainly focused on predictive models" in page 2 is not correct. The authors really need to work out what is needed and what is not. 3. Very little of the paper is really about predisposing factors. This seems to be focused on nickpoint migration, but it is it clear as to whether this is really a factor in such a large landslide. 4. The sections on the post-event landscape evolution is interesting, and probably represents the best part of the paper. But it needs to be organised in a more systematic manner that allows the reader to follow the argument. At the moment it feels somewhat chaotic and disorganised, and extremely difficult to follow. The substantive part of this section (5.2) is brief and hard to follow. There are results elsewhere though, which is confusing 5. I am not sure why so much detail is needed on the large-scale geomorphic evolution of the area. This would be better dealt with through references. 6. The discussion is also hard to follow, needing a restructure. 7. I am not sure that the review of previous studies of this landslide really present them in a correct manner.

I think the abstract needs rewriting - it does not present the contents of the paper well.

I also recommend that the authors think carefully about the figures. Fig 5 for example dows not seem to really present the information being presented, figre 16 is impossible to understand, Figure 11 needs annotation, Fig 7 is too complex to understand in its current form.

I do encourage the authors to think about resubmitting a radically revised manuscript as this is a good subject. But they would be well advised to map out a new structure for the paper, in particular with a completely new section of results and discussion, and with better organised background information.

[Figure]

**ESurfD**

---

## Author Comment (AC1) · 27 Apr 2019

**in black reviewer comments**

**in red our response**

**in green changes in the manuscript**

**REV#1: (N. Roberts)**

**GREATEST ISSUES:**

i. Overall imbalance of content, with over half of the manuscript comprising background material;

We agree. We re-organized the manuscript balancing the content between background information and new data. In particular:

1) we rewrote the *Introduction* to better outline the gap of knowledge we want to fill (lack of time constraints to the Seymareh River valley evolution before and after the Seymareh Landslide occurrence, to outline the role of the geomorphic processes both as predisposing factors for MRC processes and as response to this giant gravitational instability)

2) we reduced the regional geological and geomorphological settings by 50% into a unique section: **"2 Regional geological and geomorphological framework"**.

ii. Unnecessary details and focus on some topics (numerical modeling, pre-failure creep, seismicity and hazard/risk) that take up large parts of the abstract and/or introduction, but that are not part of the current contribution and are not directly relevant to its conclusions;

We agree. We eliminated the speculative parts of the manuscript and focused on the topics relevant for our original contribution, both in the Abstract and Introduction.

iii. Incomplete review of previous work on the Seymareh landslide and geology of Kabir Kuh, including misattribution of several findings and interpretations;

We are grateful for evidencing such a weak point and we agree. We implemented the lacking references regarding the regional geological stratigraphy and the Seymareh Landslide, attributing the information correctly.

iv. Insufficient methodologic descriptions that prevent the new work from being properly evaluated or replicated;

We agree. We implemented a complete and accurate description of the OSL technique as well as of the source and scale of the aerial and satellite imagery used.

v. Confusing organization that includes: a section purported to present new material (section 4) comprising largely a repeat of what was already known; a results section (section 5.2) containing a mix of observations/interpretations and results; and a discussion (section 6) presenting apparently new observations/interpretations.

We totally agree. We changed the structure of the manuscript, hopefully making it clearer and more understandable in its sections. We worked a lot on both the Results and Discussion in order to fix these weaknesses. In particular:

1) we eliminated the section **"4 Revised stratigraphic column and geological sections of Seymareh river valley"**

2) we removed the interpretations from the Results

3) we moved the text with new observations from the Discussion into the Results

**MAJOR ISSUES**

1. The writing is very wordy and convoluted. Many sentences are unnecessarily long and complex, making them hard to follow. Grammatical and language errors are abundant. The scale of these problems make substantial rewriting necessary, and thus in my view is a major issue. Many examples of such instances are provided in the list of minor issues below, but this is not an exhaustive list.

1. We agree.

1. The revised manuscript underwent an official language editing service (see the attached Certificate).

2. The title does not accurately convey the main thrust of the paper. It suggests that the paper provides new understanding of the causes of the landslide, which it does not. This paper is about valley-bottom geomorphic evolution before and after a gigantic landslide.

2. We agree: the focus of the paper is on the valley-bottom geomorphic evolution before and after a gigantic landslide, while the rest are implications.

2. The new proposed title is:

**"New constraints to river valley evolution before and after the emplacement of the largest landslide on the exposed Earth surface: the Seymareh rockslide - debris avalanche (Zagros Mts., Iran)".**

3. The abstract does not summarize the present study well, and instead mentions all sorts of things that are not part of the authors' work; although some of these are peripherally related (hazard, seismic triggering, causes of the landslide) they do

not constitute anything new as far as I can tell from the presentation of the rest of the paper. At the same time, the abstract lacks details about some of the major interpretations and results from the body of the paper. It needs to be rewritten and streamlined.

3. We agree: we removed from the text of the abstract the peripherally related arguments, focusing on the main topic of the valley-bottom geomorphic evolution before and after the gigantic landslide and on the methods to reconstruct it.

3. The new proposed abstract is:

**"The Seymareh Landslide detached ~10 ka from the northeastern flank of the Kabir-kuh fold (Zagros Mts., Iran), is recognized worldwide as the largest rock slope failure (44 Gm$^3$) ever recorded on the exposed Earth surface. Detailed studies have been performed that have described the landslide mechanism and different scenarios have been proposed for explaining the induced changes in landscape. The purpose of this study is to provide still missing time constraints to the evolution of the Seymareh River valley, before and after the emplacement of the Seymareh Landslide, to highlight the role of geomorphic processes both as predisposing factors and as response to the landslide debris emplacement.**

**We used optically stimulated luminescence (OSL) to date lacustrine and fluvial terrace sediments, whose plano-altimetric distribution has been correlated to the detectable knickpoints along the Seymareh River longitudinal profile, allowing the reconstruction of the evolutionary model of the fluvial valley. We infer that the knickpoint migration along the main river and the erosion wave propagated upstream through the whole drainage network caused the stress release and the ultimate failure of the rock mass involved in the landslide. We estimated that the stress release activated a Mass Rock Creep (MRC) process with gravity-driven deformation processes occurring over an elapsed time-to-failure on the order of $10^2$ ky. We estimated also that the Seymareh damming lake persisted for ~3500 years before starting to empty ~6.6 ka due to lake overflow. A sedimentation rate of 10 mm y$^{-1}$ was estimated for the lacustrine deposits, which increased up to 17 mm y$^{-1}$ during the early stage of lake emptying due to the increased sediment yield from the lake tributaries. We calculated an erosion rate of 1.8 cm y$^{-1}$ since the beginning of the landslide cut by Seymareh River, which propagated through the drainage system up to the landslide source area.**

**The evolutionary model of the Seymareh River valley can provide the necessary constraints for future stress-strain numerical modeling of the landslide slope to reproduce the MRC and demonstrate the possible role of seismic forcing in anticipating the time-to-failure for such an end-member case study."**

4. The summary of previous work on the Seymareh landslide is missing many key points and attributes some details to the wrong sources. Other details are not attributed at all. For instance:

A. Page 3, line 6: Some important and very relevant contributions of Roberts (2008) and Roberts and Evans (2013) are not mentioned. Those sources propose a detailed model of how the geologic and tectonic evolution of Kabir Kuh predisposed the slope to such large-scale failure, including formation of structural/kinematic and rheological control. As far as I can see, this

contribution is not recognized in the current paper, despite it being directly related to the authors' claimed contribution of improved understanding of factors predisposing the slope to gigantic failure.

B. Page 3, line 9: It is not sufficiently clear that the age estimate of 9800 radiocarbon years is based on the interpretation of three separate radiocarbon ages. This 9800 a BP age is taken from Roberts and Evans (2013) and must be cited accordingly. The ages provided by other sources - at least those from Griffiths et al. (2001), which were not influence by the 'hard-water effect' – should also be noted so that the reader does not have to refer back to Roberts and Evans (2013).

C. Page 3, line 6: Yamani et al. (2012) provide no new details on emplacement mechanisms of the landslides (at least not from the details in the English language extended abstract of their paper writing in Farsi). If there are some details missing from the Yamani et al.'s English text that they present manuscript refers to, it would be very helpful to provide translated quotes in the supplemental material. Otherwise, mention of new details on landslide emplacement attributed to that source needs to be removed. Yamani et al.'s (2012) main contribution comprises some general details on the evolution of lake drainage.

4. Of course, we did not mean to forget the important and very relevant contributions by Roberts (2008) and Roberts and Evans (2013), which we referred to maybe not enough explicitly. Therefore:

4A. We better referred to Roberts (2008) and Roberts and Evans (2013):

4A. **"Roberts (2008) and Roberts and Evans (2013) provided a detailed model of how the geological and tectonic evolution of the Kabir-kuh fold predisposed the slope to such a large-scale failure, including formation of structural/kinematic and rheological control, and inferred a seismic trigger."**

4B. We clarified that the Seymareh landslide age estimate of 9800 radiocarbon years is based on the interpretation of three separate radiocarbon ages based on other sources:

4B. **"Specifically, Roberts and Evans (2013) obtained from a charcoal-rich layer approximately 15 m above the base of the lacustrine sequence with a $^{14}C$ age of 8710 years BP. Based on the interpretation of three separate radiocarbon ages provided additionally by Griffiths et al. (2001) an estimated radiocarbon bracket age of the Seymareh event was suggested between 9800–8710 $^{14}C$ years BP"**

4C. We corrected the text describing the Yamani et al. (2012) main contribution.

4C. **"Yamani et al. (2012) provided some general details on the evolution of the dam lake drainage, describing a sequence of entrenched lacustrine terraces upstream of the landslide dam."**

5. The paper includes a lot of largely unimportant, or at least overly specific, background details. Many of the details about tectonic features and some of the details about seismicity in section 2 are well beyond what is necessary to provide relevant background to the reader. These extraneous details could appear in the supplement to provide further context for the interested reader, but they take up too much of the main paper. Given that modeling is not part of the present paper, much of the background provided about modeling is irrelevant.

5. We agree: we merged the Section 2, Section 3 and the Section 4 removing many of the details about tectonic features and some of the details about seismicity and about the modelling that are included in the supplementary material.

5. The new background section named "**2 Regional geological and geomorphological framework**", is as following:

[revised manuscript text omitted]

6. If the mechanism and behaviour of large bedrock landslides are to be discussed, some recent reviews pertaining to this topic should be cited (principally Brideau and Roberts [2015], which includes this event as a case study, and Hermanns and Longva [2012]). Those sources provide numerous additional references on progressive failure that should also be considered if the authors can make a suitable case for discussing this topic.

6. In fact, the mechanism and the behavior of large bedrock landslides are not part of the current contribution and are not directly relevant to our conclusions. Purpose of the study is rather to provide still missing time constraints to the evolution of the Seymareh River valley, before and after the emplacement of the Seymareh Landslide, to highlight the role of geomorphic

processes both as predisposing factors and as response to massive rock slope failures. Therefore, we decided not to include the suggested references in the manuscript.

7. Much of the background information lack proper referencing, including tectonic setting (section 2), seismicity (section 2), and geologic setting (section 3). For example, the paragraph starting on page 4, line 27 presents many details that are clearly not part of the current study, but only provides references in two places. Furthermore, this generalization of sources does not allow the reader to sort out what details have come from what sources. The hypothesis cited in the final paragraph of section 2 is based on much more than just the frequency of strong earthquakes in the region, as this submission suggests. Progressive steepening of the slope at a very slow rate relative to the modern recurrence rate of nearby strongly felt earthquakes is a crucial consideration as it makes failure initiation in the absence of seismic loading hard to explain.

7. We agree. First, the background information including tectonic setting, seismicity and geologic setting were strongly reduced and many of the details about tectonic features and seismicity were included in the supplementary material.

7. The new background section "**2 Regional geological and geomorphological framework**") is attached in the point 5. More specifically, the paragraph regarding the seismicity is reduced to the following:

**"Seismicity is distributed in a 200-300 km wide area of the Zagros mountain range (Hatzfeld et al., 2010, Paul et al., 2010, Rajabi et al., 2011), with a sharp cut along the Main Zagros Reverse Fault in the NE (e.g., Yamini-Fard et al., 2016), with recurrent earthquakes of Mw 5-6 and exceptional earthquakes of higher magnitude, i.e., up to Mw 6-8 (see supplementary material). The SL occurred in a very densely seismically active area and recurrence rate of nearby strongly felt earthquakes considerably higher than the rate of slope steepening led Roberts and Evans (2013) to hypothesize that seismic forcing may have played a primary role in triggering the landslide."**

8. Thickening of the Pabdeh Formation (top of page 6) very well may have in influence on landscape development. However, the position of the nw trending deformation front also strongly inferences the westward change in landscape. This was also recognized by Oberlander in his work but appears to have been overlooked.

8. We strongly reduced the regional geomorphological setting, including it in a unique background section, named "Regional geological and geomorphological framework", by removing parts not functional to the results presented, included the text this comment is referred to.

8. The paragraph regarding Oberlander's landscape evolution model is the following:

**"Oberlander (1968) suggested that the drainage network in the NW Zagros was superimposed from structurally conformable younger horizons. In his model, the breaching of hard geological units of the antiformal ridges follows a phase of river cutting and expansion of the fold axial fold basins through the softer overlying units. In the Kabir-kuh fold, the transverse cutting of the Asmari limestone, and the exposure of the underlying more erodible Pabdeh-Gurpi marls, leads to the formation of a low-relief landscape with synformal ridges on which the new through-going drainage system can be developed. In Oberlander's hypothesis, it is the Pabdeh and Gurpi marls that facilitate the**

**creation of a low-relief landscape across the anticline crests and are therefore integral to the story of drainage superimposition.”**

9. In contrast to what the authors claim, I see no evidence that this work revised the the stratigraphy of the study area to any substantial degree. The stratigraphy of the part of this Zagros fold-thrust belt is extensively reported in previous work (much of which has not been referenced here). The stratigraphy described here matches very closely (in sequence order, composition, and thickness) with the stratigraphy already reported in the literature. For instance, I see very little difference between the sequence in Fig. 5 of the present manuscript and the stratigraphy summarized from review of existing literature by Roberts and Evans (2013, their Fig. 5). Several important references on the sequence are missing and should be included. For the overall region this includes James and Whynd (1965) and Alavi (2004). Detailed mapping of Kabir Kub conducted by Iran Oil Operating Companies (Setudehnia and Perry, 1967; Takin et al., 1970; Macleod, 1970) already covers much of what the authors would have covered in their ’new’ mapping presented in section 4 and in Fig. 5. The authors need to first clearly describe what has already been documented (in a section on geologic background). Only after that should they try to justify how their ’new’ stratigraphy differs. From what I can tell, they contribute only some additional detail on the Pabdeh Formation, although at least some of this is similar to that reported in the Iran Oil Operating Companies maps. A much more convincing argument will need to be made if any new contribution to the area’s stratigraphy/geology is to be claimed. The details presented in section 4 nearly all belong in the background material. This further highlights an issue with the paper’s layout and balance: the new contribution of the 15-page manuscript doesn’t start until page 9, meaning that over half of the paper is background information.

9. This was effectively a mistake, induced by the detailed field work we performed. Such a field-work included also a deep check of the local stratigraphic column, which in turn did not provide any significant new data with respect to the literature. We re-organized the text by removing Section 4 and integrating the text within the unique background section named “Regional geological and geomorphological framework”.

9. **“The outcropping formations in the Kabir-kuh anticline date to a time interval ranging from the Late Cretaceous to the early Miocene and are characterized by different lithological and rheological properties (Vergés et al., 2011). Since the geo-structural setting of the fold flanks represented a crucial predisposing factor for the catastrophic massive rock slope failure (Roberts and Evans, 2013), we referred to the most detailed stratigraphic column proposed by James and Wynd (1965), Alavi (2004) and to the detailed mapping of the Kabir-kuh fold conducted by the Iran Oil Operating Companies (Setudehnia and Perry 1967; Takin et al. 1970; Macleod 1970). Specifically, the investigated area includes the middle and low reaches of Seymareh River starting approximately 60 km upstream of the SL down to the SE termination of the Kabir-kuh fold. In Fig. 2, the geological map of the study area, the stratigraphic column and two geological cross-sections related to different structural sectors are reported.**
**It is noteworthy that, in the Kabir-kuh anticline, the Pabdeh Formation is composed of three rheologically contrasting members, which crop out in the SL scar area: i) the lower Pabdeh member (150 m thick), which is**

**dominated by marls and shales, ii) the Taleh Zang member (50 m thick), consisting of platform limestone, and iii) the upper Pabdeh member (150 m thick), composed mainly of calcareous marl. The Asmari Formation creates a carapace originally covering the top of the Kabir-kuh fold, while in the synclinal valleys between the Kabir-kuh fold and the adjacent folds, the Asmari Formation is overlapped by a Miocene-Pliocene succession (Homke et al., 2004).**

5  **Referring to the Changuleh syncline studied by Homke et al., 2004, the foreland stratigraphy includes the following: i) the Gachsaran Formation (early Miocene - 12.3 Ma, thickness approximately 400 m), composed of salt, anhydrite, marl and gypsum; ii) the Agha Jari Formation  (12.3 Ma – 3 Ma,, thickness approximately 1400 m); and iii) the Bakhtiari Formation (3 Ma – early Pleistocene, thickness approximately 900 m). The Agha Jari Formation consists of sandstones and conglomerates, linked to the evolution from deltaic to fluvial transitional environments (Elyasi et al.,**

10  **2014), and the Bakhtiari formation consists of conglomerates characterized by coarse and mud-supported grains, sandstones, shales and silts and marks the onset of syn-orogenic fluvial environment conditions (Shafiei and Dusseault, 2008)."**

10. The Gachsaran formation has a high gypsum content, which dominates its geomechanic and geomorphic behaviour. The

15  description provided here (Page 8, line 17) instead suggests that it comprises only more typical clastics. Given this manuscript's focus on valley-bottom evolution, this unit needs to be accurately characterized.

10. This was a typo (Gachsaran instead of Agha Jari) that we corrected.

10. **"The Agha Jari Formation consists of sandstones and conglomerates, linked to the evolution from deltaic to fluvial transitional environments (Elyasi et al., 2014)"**

11. The details, or even relevance, of anticline flank dips (second paragraph of section 4) are unclear. Dips along the three sections probably need to be considered more carefully. Due the nature of the anticline, structural variation is to be expected. Dips will of course decrease to the southeast toward the nose of the anticline. What is possibly more interesting is how much steeper dips are to the northwest beyond the mapping presented here. Furthermore, this is a complex box fold with a hinge

25  near the upper part of the Seymareh detachment zone. Downslope variation flank dips will reflect this structure. Due to the complexity of the fold, stratigraphic position will also affect dip. For example, Roberts and Evans (2013) noted steepening (see their Fig. 9H for example) of the Eman Hassan member in the upper, central part of the landslide scar. In contrast, the adjacent upper surface of the Asmari Formation on either side of the landslide is much less steep. Variations between other units are noted also in mapping by Iran Oil Operating Companies (summarized in part in Fig. 4B of Roberts and Evans).

30  Such variation has important implications for dip-slope failure of the flank. In light of these points, it would thus be very helpful (and far more informative) if the authors provided a detailed map of their structural measurements and clarify how these may build upon those provided by past studies of the landslide and by mapping by Iran Oil Operating Companies and by Roberts and Evans (2013). Conversely, if these measurements are based only DEM profiles, that needs to be clearly stated. In any case, how well profiles through the eroded core of the anticline represent protections the Asmari limestone

limbs beneath the valley floor needs to be evaluated given stratigraphic variation in dips relating to the complex nature of the flood. Finally, profile C-C' is oblique to the true dip of the anticline flank and appears to thus under represent dip of the Asmari surface (based on Fig. 5, it appears that the apparent dip, not the true dip, is being represented).

11. Since we do not provide in this work any new data on the geological structural setting of the fold flank, we moved to the new background section "Regional geological and geomorphological framework" the geological map and sections and just recall in the Discussion the arguments already provided by Roberts and Evans (2013). In this perspective, the profile C-C' was eliminated, since not necessary.

11. **"The reported cross-sections intersect the synclinal valley of the Seymareh River. The dip angle of the northeastern flank of the syncline considerably decreases from NW to SE from 45° (section A-A') to 18-20° (section B-B'). Therefore, along the section A-A' the Cretaceous-Paleogene bedrock (from the Sarvak Formation to the Asmari Formation) offers a greater accommodation volume to the continental and epicontinental formations (Gachsaran Formation and Agha Jari Formation), as the synclinal axis is located at a lower elevation than in the B-B' section."**

12. Is this the reconstrunction of the Asmari carapace (Page 8, line 29) immediately pre-failure or a reconstruction of the anticline prior to unroofing? Please clarify. In the latter case, 2100 m a.s.l. is a substantial underestimate and the structure of the box fold suggests that the Asmari carapace extended much higher.

12. In the light of the response above (Point 11) this point is no longer discussed.

12. See the changes in the manuscript attached at point 11.

13. The authors' point about the position of the Seymareh River (Page 8 line 31) does not become clearly relevant later one in the paper. Depending on its significance for the current study, which remains unclear, the authors will need to clarify whether this river position is related to migration of the channel within the Lake Seymareh lacustrine deposits (i.e. following valley damming and sediment in-filling of the Lake Seymareh basin) or is the result of some older physiographic control. The river has obviously migrated to the northwest in the last several hundred years as a Sasanian-era bridge over the old river position at Pul-i-Shikari is nowhere near the modern river (see observation and discussion by Harrison and Falcon, 1938).

13. This point is no longer discussed since we removed the entire Section 4. Nonetheless, we recognized in the field relict landforms that testify for the presence of an abandoned valley (that we attributed to a paleo-Seymareh river), which in important in our Evolutionary model of the landscape. The northwestward migration of the river in the last several hundred years, is not related to the long-term evolution of the valley that we discuss, but seems attributable to 'physiological' local variations of the river path, which could fall in the normal variability, of a meandering path in a short time-scale.

13. See the changes in the manuscript attached at point 11.

14. I cannot understand what the final five lines of section 4 mean. I assumed the kinematic release suggested here is that of the Seymareh landslide. What is the 'connectivity' supposed to be? How are the flatirons envisioned to control sliding? Note that several previous studies suggest fluvial undercutting of the slope as the source of kinematic freedom at the slope toe. However, breakout across the upper units is also necessary for the failure to have occurred as the failure surface cuts

5  stratigraphically upward in the downslope direction (Roberts and Evans, 2013); this feature is an important part of the kinematic release.

14. This point is no longer discussed since this part was removed with Section 4.

14. See the changes in the manuscript attached at point 11.

10  15. The methods section lacks sufficient detail. What are the source and scale of the air photos used? Were they interpreted quantitatively or only qualitatively? What specific imagery was used from Google Earth (there is of course a very wide range of imagery types and qualities available in that software)? What as the source of the map used? What inputs specifically were used the creating the DEM (current wording is unclear)? No methods are provided for the new geologic investigation that the authors claim to have conducted. OSL can be a finicky technique. Many critical aspects of the sampling are not

15  considered, particularly those necessary to rule out partial bleaching during sample collection and transport: was the slope cleared off first? was an opaque sample vessel used? how far was it inserted into the slope? OSL sampling methodologies vary quite a bit, so unless the approach exactly follows that of a pervious study, simply citing in past source here is insufficient.

15. We agree: we implemented the lacking information.

20  15. Attached the new **"3 Methods"** section:

[revised manuscript text omitted]

16. the results section (section 5.2) is rather hard to follow and contains a mix of observations, interpretations and results. furthermore, several observations/interpretations (conveyed in Figs. 14 and 15) are skipped over here and are referenced only in the discussion. The formatting of this section needs work. I see no benefit to using lists here (or anywhere in the paper); all text should be in paragraph format. Several parts of the lists are not even full sentences. Issues with in the content include a lack of clarity over how the authors believe the dated sequences to relate to each other. For example, the wording at the start of the section (Page 10, line 9) seems to suggest that the lacustrine terrace pre-dates the landslide, but I cannot imagine how this is possible. Such as suggestion also conflicts with the post-landslide age reported for samples SEY4 and SEY5.

16. We agree: we reduced and formatted a new more concise results section without interpretations.

16. Attached the new **"4 Results"** section:

[revised manuscript text omitted]

10 17. The discussion lacks any details about reliability of osl ages presented here. There are many possible error sources in this technique, but it is unclear if these were considered.

17. The error sources of OSL ages are discussed in depth in the new Method Section (see point 15) and the degree of error of the ages is reported in the table as a column (see Table 1). Talking about the argument also in the Discussion Section could be quite redundant.

18. The discussion suggests that the current study provides some new insight on the geologic succession and its role in mass rock creek (MRC). I see no such contribution in the paper. the authors seeming claim in the abstract and introduction that their study somehow addresses pre-failure creep, but this topic is not investigated in any detail. Pre-failure creep is hardly mentioned, and even then, is based only a couple of field observations. The suggestion that features noted in the Pabdeh
20 Formation (Page 13, line 2) indicate pre-failure creep is not sufficiently supported. It is also possible that these features are not a result of progressive failure of the slope. The plastic deformation shown in Fig. 14B could well be the result of pre-failure creep, but not enough detail is provided to evaluate this. Rock mass strength reduction and associated deformation is also to be expected as a result of fold formation (see Roberts and Evans, 2013 and references therein). The brittle deformation shown in Fig. 14C could well be the results of sliding during catastrophic failure, so I see no reason to use it to
25 argue for pre-failure.

18. Although we agree that the current study does not provide any new insight on the geologic succession, we disagree on the remaining content of this comment: it is important to note that based on the known stratigraphy, a rheological contrast in terms of stiffness and viscosity can be hypothesised between Asmari Formation, the Upper Pabdeh and the Taleh Zang members. Evidence of ductile deformations within the Pabdeh layers, which cannot be ascribed to parasitic structural folding
30 (since the can be only observed just below the SL sliding surface), and buckling landforms that demonstrate relevant rebound effects (which are related to an intense stress release after the SL collapse) can be regarded as a proxy for MRC affecting the SL slope before failure due to gravity-induced deformations. Much of the literature ignores that this process is the basis of the triggering of this landslide, while in our opinion this is a fundamental point. In order to better explain this

point, not only we described the above cited field evidence in the Results, but we also added a specific sub-section in the Discussion which is **"5.4 Implications of the evolutionary model for future back-analysis of the SL".**

18. **"According to the multi-modeling approach proposed by Martino et al. (2017), Quaternary landscape evolution modeling of slope-to-valley floor systems plays a key role as a tool for chronological constraints to the creep evolution of entire slopes (Bozzano et al., 2016; Della Seta et al., 2017).**

**The geomorphic processes developed before the failure of the SL likely acted as predisposing factors for MRC processes in the rock mass successively collapsed. Kinematic freedom, both at the top and on the fold flank was created by the incising network of streams that dissect the Asmari Formation carbonate caprock following the major joint set in the Asmari Formation already described in Roberts and Evans (2013; and references therein). In particular, the headward erosion of streams towards the anticline's structural high described by Oberlander (1968), caused the expansion of the fold axial basins through the softer units, determining the upslope kinematic freedom. In the timing proposed by Tucker and Slingerland (1996) the latter was reached at approximately 1.6 ka. Stress release at the slope base was definitely produced by the Middle-Late Pleistocene upstream migration of the knickpoint along the Seymareh River longitudinal profile. Unfortunately, since the emplacement of the landslide swept away the uppermost outcrops of the alluvial terraces formed in response to the upstream knickpoint migration, the rate of knickpoint migration cannot be inferred. Nonetheless, an elapsed time-to-failure on the order of $10^2$ ky, since the kinematic freedom at the slope base was reached, can be reasonably estimated by the age of the oldest terrace in the lower reach of the river minus the age of the landslide occurrence.**

**It is noteworthy that the stratigraphy of the source rock mass, also described in detail by Roberts and Evans (2013), accounts for different rheological behaviors, which could have induced differential strain rates within the slope leading to failure according to a MRC process. More particularly, the time-dependent visco-plastic behavior, more typical of clayey and marly deposits, which have lower viscosity values, can justify time-dependent (creep) strains which could have generated high stress concentration within the higher viscosity level over time (i.e., mostly characterized by elasto-plastic rheology), inducing their cracking and leading to failure. In fact, a stiffness contrast exists between the upper member of the Pabdeh Formation and the overlying Asmari Formation. The attitude of the strata is moderately dipping downslope (15°-20°), and a reduced lateral confining effect is due to continental and epicontinental deposits ascribable to the Gachsaran and Agha Jari Formations. Moreover, the low dip angle of the strata reduces the vertical thickness of the Asmari Formation caprock, which was completely eroded by Seymareh River during its engraving, thus allowing the sliding mechanism of the Pabdeh and Asmari layered formations.**

**Therefore, the results of this work have implications for a future back-analysis through stress-strain numerical modeling of the SL because they can be used to constrain the elapsed time since MRC initiation and ultimate failure conditions. Such a perspective is to be regarded as a key challenge for dimensioning such an end member event in regard to both time and space distribution as well as for evaluating the possible role of impulsive triggering actions (i.e., strong to very strong earthquakes) in anticipating the time-to-failure of the slope."**

19. The authors also bring up numerical modeling in the abstract, introduction and discussion, despite it not factoring into the study. Perhaps they mean to indicate that their results could be used to guide/inform numerical modeling, but their wording is unclear. Given that numerical modeling is not addressed in the study, the background on modeling in the introduction is too extensive and Figure 14 seems irrelevant.

19. We agree that the numerical modeling was not performed in this work. Nonetheless, the evolutionary model of the river valley here proposed provide useful constrains for future stress-strain numerical modelling of the landslide slope to reproduce the MRC and the possible role of seismic forcing in anticipating the time-to-failure of the gravity-driven deforming processes for such an end-member case study of rock slope failure. We better explained these implications in the text in a specific section which is **"5.4 Implications of the evolutionary model for future back-analysis of the SL".**

19. See the changes reported at point 18.

20. From what I can tell, the authors provide no clear new evidence regarding predisposure of the slope to failure. They start to suggest two interesting aspects, but do not adequately address them. The first aspect is that of pre-failure creep (noted above). The second, which is more closely tied to their investigation, is the possibility of knickpoint migration along Seymareh River, which is inadequately communicated. The extensive evaluation geologic and geomorphic controls on the failure by previous studies (particularly Roberts and Evans, 2013) is hardly addressed, although authors of the present study claim that this is one of their main focuses.

20. We totally agree: we discussed more in depth these points about the pre-failure valley evolution in the Discussion, both in "**5.1 Constraints to pre-failure valley evolution**" and in **"5.4 Implications of the evolutionary model for future back-analysis of the SL"**

20. Attached the "Constraints to pre-failure valley evolution" section**:**

**"The longitudinal profile of the Seymareh River and the geomorphic markers preserved mainly downstream of the landslide dam provided new constraints on the pre-failure valley evolution. The major knickpoint located immediately upstream of the SL is the most interesting to be analyzed in relation with the landslide event. Its shape in the longitudinal profile clearly let us identify it as a "slope-break knickpoint" (Kirby and Whipple, 2012; Boulton et al., 2014), thus developed as a knickpoint retreating in response to a persistent perturbation to the fluvial system (Tucker and Whipple, 2002), as frequently observed in tectonically active regions. The location of this knickpoint upstream of the SL and the outcrop of the basal contact of the landslide at the bottom of the Seymareh River gorge (Fig. 7a) suggests that this shape of the longitudinal profile was already developed before the failure, meaning that the erosion wave which generated the knickpoint affected the SL slope foot before the failure occurrence.**

**The poorly preserved, well-cemented alluvial fan conglomeratic deposits outcropping upstream of the landslide lie on the Miocene Agha Jari Formation, at a higher elevation than the outcrops of the Bakhtiari Formation. Their remnants are aligned in correspondence with the axis of a relict synclinal valley, likely corresponding to a very early**

**stage (Pliocene?) of the Seymareh valley evolution. On the other hand, the conglomerate deposits outcropping downstream of the landslide (Cg_l) are closer in height to the major knickpoint, thus suggesting that they were in equilibrium with a local base level corresponding to the early propagation of the major knickpoint. Furthermore, they must be younger than the Bakhtiari Formation, which is preserved at higher elevation.**

**The alluvial terraces preserved downstream of the SL likely mark the valley evolutionary stages during the major knickpoint retreat (Demoulin et al., 2017). Along the longitudinal river profile, the uppermost outcrops of each level of this terrace suite were swept away by the landslide, which unfortunately prevents estimation of the rates of knickpoint retreat. Nonetheless, according to what was observed by Bridgland et al. (2017) about river terrace development in the NE Mediterranean region, the sedimentation phases should correspond to cold periods. In particular, Bridgland et al. (2012) observed, in the valleys of the Tigris and Ceyhan in Turkey, the Kebir in Syria and the trans-border rivers Orontes and Euphrates, a regular terrace formation in synchrony with 100 ka climatic cycles that can be correlated with MIS 12, 10, 8, 6 and 4-2. Therefore, the minimum ages obtained for the SEY3 and SEY10 samples could be reasonably extended to 478 ka (MIS 12) and 374 ka (MIS 10), respectively, and the OSL age of the SEY11 fits well with the Last Glacial Period."**

See also the changes reported at point 18 for **"5.4 Implications of the evolutionary model for future back-analysis of the SL"** section**.**

21. Why are landslide kinematics described in the discussion (top of page 13) when they are not part of the new work presented here and when the detailed examination of this topics by previous workers is not mentioned in the background sections? For instance, the geomechanical strength contrast between the Asmari and Pabdeh formations is mentioned in the current paper but has not been characterized. It is, however, approximated in Roberts and Evans (2013). What evidence is there for kinematic freedom provided by gullies along the flank of Kabir Kuh (Page 13, line 13)? The lateral margins of the main landslide are nearly vertical features following a major joint set in the Asmari Formation (characterized in Roberts and Evans, 2013 and references therein, but not mentioned here). There is no evidence I can see for these being related to fluvial processes. Roberts and Evans (2013) propose that these features are instead inherited from the tectonic history of the Zagros' simply folded zone.

21. In the new structure of the manuscript we better explain the role of the drainage network in the onset of the kinematic freedom of the rock mass in the subsection **"5.4 Implications of the evolutionary model for future back-analysis of the SL"**.

21. See the changes reports at point 18.

22. What is the basis for the suggestion that failure was preceded by an 'elapsing time' on the order of 100 ka (Page 13, line 17)? It's not even clear what this period is meant to represent. Is this a period of pre-failure creep? The period between the knickpoint passing the toe and the slope failure? This is very hard to follow.

22. See the above response (point 21).

22. See the changes reports at point 18.

23. The conclusion has several issues. It seems to include material – some kind of modeling related to the Seymareh River valley – that is not only not part of the present study, but as far as I can tell is not included in any other published research. I do not see how this fits in, other than also being mentioned in the abstract and introduction. The list summarizing landscape evolution of this part of the Seymareh River valley seems very similar to the list presented on the previous page in the in the discussion and is thus quite redundant. The conclusion ends with a sentence about seismic triggering, which is hardly mentioned in the paper other than stating that pervious workers have suggested it (and even then fails to adequately explain what has been done before). I don't see how the new work done here will contribute to evaluation of a seismic trigger. This text seems to be irrelevant to the conclusions of the paper.

23. We agree: we removed the list presented in previous section and focused on the main topic resulted from the study.

23. Attached the revised conclusion section:

**" In a multi-modeling approach to the study of MRC processes affecting slopes at a large space-time scale, the performed geomorphic analysis allowed us to constrain the evolution of the Seymareh River valley in the northwestern Zagros Mts., before and after the failure of the largest landslide ever recorded on the exposed Earth surface. The identification and OSL dating of different suites of lacustrine, alluvial and strath terraces constrained in time the major pre- and post-failure evolutionary steps of the river valley system.**

**The oldest geomorphic markers in the Seymareh River valley are represented by relict conglomerates preserved upstream of the landslide, which demonstrate the early (Pliocene?) position of a paleo-Seymareh river flowing into a synclinal valley close to the northeastern flank of the Kabir-kuh fold.**

**Drainage evolution associated with the growth of the Kabir-kuh fold was characterized by the deep incision of the stream network, which allowed the kinematic release of the rock mass involved in the Seymareh giant landslide. Such a stream incision was accompanied by the retreat of a major "slope-break knickpoint" along the Seymareh longitudinal profile, time-constrained by the age of a suite of river fill terraces. According to the age of pre-failure terraces, in the middle-late Pleistocene the erosion wave reached the portion of the Kabir-kuh fold that ~10 ka was affected by the SL. According to the timing of the landscape evolution model proposed by Tucker and Slingerland (1996), the upper slope underwent kinematic release about 1.6 ka. Therefore, the collapse was prepared by MRC processes acting over a time window of $10^2$ ky;**

**The geomorphic response to the landslide dam consisted in the formation of three lakes, among which Seymareh Lake persisted for ~3500 years before its emptying phase started ~6.6 ka due to lake overflow. A sedimentation rate of 10 mm $y^{-1}$ was estimated for the lacustrine deposits, which increased up to 17 mm $y^{-1}$ during the early stage of lake emptying due to the increased sediment yield from the lake tributaries. Since ~4.5 ka, a suite of four alluvial terraces**

upstream of the landslide demonstrates the alternating erosion/deposition phases of the re-established Seymareh River.

An incision rate of 1.8 cm y$^{-1}$ was estimated since the beginning of the landslide cut by Seymareh River, and such a strong erosion started propagating up to the landslide source area where badlands developed, eroding the marly Pabdeh-Gurpi Formation.

The results obtained here provide new constraints to the valley evolution in view of future stress-strain numerical modeling of the MRC process that involved the SL slope before its generalized collapse. Such a modeling could also be considered to discuss the possible role of impulsive triggering (earthquakes) in anticipating the time-to-failure due to the gravity-driven deformational processes."

24. The figures are too numerous. Several can be combined (particularly the photos) and others seem to have limited relevance. Fig. 2 is probably more appropriate in the supplement given the lake of relevance of seismicity to the current study. Figs. 1 and 3 could probably be combined, especially if some of the extraneous detail in Fig. 1 is removed. Figures 13 and 14 are irrelevant to the main focus of the paper. Fig. 13 does not add anything to the paper. Figure 14 relates to the suggestion of prefailure creep of the slope, which contrary to what the authors state in the introduction of the paper, is not a major component of the present study. Figure 15 seems to be an afterthought, although it has far more relevance to the paper's focus on valley-bottom geomorphology that either of the preceding figures.

24. We agree: Fig. 2 was included in the supplementary material. Fig. 1 and Fig. 3 cannot be combined because they refer to very different space-time scales. Fig. 13 was included in the new Fig. 2, as a field evidence of the already known outcropping stratigraphy. While Fig. 14 (new Fig. 11) was moved to the Results section. Also Fig. 15 was moved to the Results section, being included it in the new Fig.5. Finally, Fig. 9 was integrated the Fig. 8 (new Fig. 7).

[Figure]

**Figure 2: Geological Map, stratigraphic column and cross sections of the study area.**

[Figure]

Figure 5: Geomorphic markers upstream of the SL, represented by a suite of four orders of alluvial terraces entrenched in the lacustrine deposits (Lac) of Seymareh Lake upstream of the landslide, in the areas where Harrison and Falcon (1938), Roberts and Evans (2013) and Shoaei (2014) hypothesized the natural damming lake could be extended. a) Overall view of the suite of terraces; b) example of fluvial terrace deposit; c) example of lacustrine deposit; d) evidence of a prograding lacustrine fan delta formed by one of the right tributaries of Seymareh Lake during its early emptying phase.

[Figure]

**Figure 7: Geomorphic markers upstream of the SL. a) A strath terrace and a flood plain developed over the landslide debris, which are important markers of the evolution of the natural dam since they testify to the moment of the overcoming of the damming lake and the overflooding of the river onto the landslide debris, respectively; b) detail of the strath terrace deposit sampled for OSL dating; c) The suite of fluvial terraces downstream of the SL; the Qt1 level is poorly preserved and not visible in this photo.**

The list below provides references for several works that the authors should consult and that are missing from the current paper:

Alavi, 2004. Regional stratigraphy of the Zagros fold–thrust belt of Iran and its proforeland evolution. American Journal of Science, 304, 1–20. Brideau and Roberts, 2015. Mass movements in bedrock,in: Landslides Hazards, Risks and Disasters, [Davies and Shroder, eds.]. Academic Press, Amsterdam,Netherlands, 43–90.

Griffiths et al. 2001. Environmental change in southwestern Iran: the Holocene ostracod fauna of Lake Mirabad. Holocene, 11, 757–764. Hermanns and Longva, 2012. Rapid rock-slope failures, in: Landslides: Types, Mechanisms and Modeling [Clague and Stead, eds.]. Cambridge Univ. Press, Cambridge, UK, 59–70.

James and Wynd, 1965. Stratigraphic nomenclature of Iranian Oil Consortium Agreement Area. AAPG Bulletin, 49, 2182–2245.

Macleod, 1970. Kabir Kuh, 1:100000 Geological Map. Iran Oil Operating Companies, Geological Exploration Division, Tehran.

Setudehnia and Perry, 1967. Dal Parri. 1:100000 Geological Map. Iran Oil Operating Companies, Geological Exploration Division, Tehran.

Takin, M., Akbari, Y. & Macleod, J.H. 1970. Pul-E Dukhtar. 1:100000 Geological Map. Iran Oil Operating Companies, Geological Exploration Division, Tehran.

We added these reference in the manuscript

**MINOR ISSUES**

1. Page 1, line 10: The anticline if variously referred to as 'the Kabir-kuh fold', 'the Kabir-kuh Fold', 'Kabir-kuh fold', and 'the Kabir-kuh'. Kabir-kuh is a proper physiographic feature whereas the fold feature is not officially recognized as a name. Thus, the only proper version of the naming used here are 'Kabir-kuh' and 'the Kabir-kuh fold'.

1. Right.

1. Changed.

2. Page 1, line 15: Proper capitalization and use of 'the' are: 'the Seymareh River valley' and 'Seymareh River'. This needs to be corrected throughout the manuscript.

2. Right.

2. Changed.

3. Page 2, line 28: What is meant by 'different evolutionary stages'?

3. Right.

3. Removed when re-writing.

4. Page 2, line 28: 'allows to construct' is improper language.

4. Right.

4. Changed.

10   5. Page 2, line 28: What are 'interesting valley sections'?

5. Right.

5. Removed when re-writing.

6. Page 2, line 33: The location description is incomplete. The Seymareh River valley straddels the border between Lorestan

15   and Ilam provinces, respectively to the east and west of the river. The landslide initiated in what is now Ilam province, but

most of the debris lies in Lorestan.

6. Right.

6. Removed when re-writing.

20   7. Page 2, line 34: What is meant by 'evolutionary scenarios'?

7. Right.

7. Removed when re-writing.

8. 9. Page 3, line 1: I am not familiar with the region 'External Zagros Mountains'. It this the simply-folded zone/belt?

25   8.9. Right.

8.9. Changed.

10. Page 3, line 13: 'Seimareh' should be 'Seymareh'. Although various spellings have been used over the years, 'Seymareh'

seems to be the currently recognized version. In any case, spelling should be consistent throughout the manuscript. This

30   corps up in a few other places.

10. Right.

10. Changed.

11. Page 3, line 15: It is far more informative to state here what the study achieved, rather than what it intended to achieve. Also 'aims at better understanding' is improper English.

11. Right.

11. Changed.

12. Page 3, line 16: Here, in the abstract and again later on risk (or risk mitigation) is thrown in. However, this topic is not explored. Practically, the only mitigation would be complete evacuation (either temporality based on some kind of waring system or permanently) of an area that could experience landslide of this magnitude and stabilization or localized avoidance are impossible. Furthermore, the very low probability of a landslide of this magnitude means that it risk is potentially rather

10 low.

12. Right.

12. Removed when re-writing.

13. Page 3, line 25: What is 'e.g.' used? Also, this is hardly the more relevant source for this statement given that several

15 studies have investigated the landslide.

13. Right.

13. Removed when re-writing.

14. Page 3, line 27: Units should be m a.s.l. as this is an elevation.

20 14. Right.

14. Changed.

15. Page 4, line 1: 'The Zagros' is not the proper way to refer to the range. This statement also requires references to back it up.

25 11. Right.

11. Changed.

16. Page 4, line 15: 'landslide' not 'Landslide'.

11. Right.

30 11. Changed. When we refer to the Seymareh Landslide, we use SL, as in Shoaei (2014).

17. Page 4, line 15: Given that this is a new paragraph, reference back to the previous content (using 'latter') is a confusing.

17. Right.

17. Changed.

18. Page 4, line 22: Parentheses missing around publication year.

18. Right.

18. Changed.

19. Page 4, line 22: What is meant by the 'onset of the [sic] deformation'? Is this supposed to be propagation of the deformation front?

19. Right.

19. Removed when re-writing.

20. Page 5, line 13: I've never come across 'Delful Zagros' as a term. Are the authors certain that this is a properly recognized physiographic region?

20. Right.

20. Changed.

21. Page 5, line 14: Missing year of source.

21. Right.

21. Changed.

20  22. Page 5, line 21: The mobile and competent units have not yet been introduced, and are part of the geology not the geomorphology (as the section title would suggest). These have not yet been introduced. The geology should be briefly summarized before the geomorphology, especially given the apparent influence of the former on the latter. A few of the units are mentioned in the following lines, but the geology is of course much more complex than that.

22. Right.

25  22. Changed, including in the new "**2 Regional Geological and Geomorphological Framework**" the description of the stratigraphy.

23. Page 5, line 22: This fold is in Ilam province, not Lorestan. The boarder between them in this area follows Seymareh River.

30  23. Right.

23. Changed.

24. Page 6, line 28: Should be 'the Asmari Formation'. Spell formation out throughout the text of the paper; do not abbreviate to 'Fm.' (as on line 31).

24. Right.

24. Changed.

25. Page 6, line 5: This is general physiographic background that should appear much earlier on in the paper.

25. We completely re-wrote the Regional Geological and Geomorphological Framework, deleting the unnecessary text.

26. Page 6, line 6: Identify the lakes here and how they were formed (i.e. which rivers were blocked). Lake Balmak is not named until the discussion and is hard to place in the figures. It would also be helpful to very briefly note that much of the previous literature calls this lake Chah Javal.

26. Right.

26. Changed.

27. Page 6, line 9: Regarding '. . .formed in response to a sequence of landslide', clarify whether this is multiple separate landslides or all related to the Seymareh landslide.

27. Right.

27. Changed.

28. Page 6, line 9: Consider simplifying to 'the landslide dams' so that the reader does not mistake your meaning to be multiple landslide dams of Seymareh Lake (instead of multiple lakes dammed by the Seymareh landslide).

28. Right.

28. Changed.

29. Page 7, line 12: Combining 'none' and 'neither' forms a double-negative. Also, 'study' should be 'studies'.

29. Right.

29. Changed.

30. Page 7, line 13: Specify 'fluvial' geomorphic markers (and remove 'the').

30. Right.

30. Changed.

31. Page 7, line 17: 'landslide' not 'Landslide'.

31. Right.

31. Changed. When we refer to the Seymareh Landslide, we use SL, as in Shoaei (2014).

32. Page 7, line 18: Presumably 'refer to' means 'date to'?

32. Right.

32. Changed.

33. Page 7, line 19: Geologic ages should be late/early, not upper/lower. The latter pertain the stratigraphy, not ages.

33. Right.

33. Changed.

34. Page 7, line 19: The ages should be 'Late Cretaceous' (an officially recognized age) and 'early Miocene' (not an officially recognised age).

34. Right.

34. Changed.

35. Page 8, line 21: Unless I've missed a break, the paragraph ending on this line is massive and needs to be broken up.

35. Right.

35. Changed.

36. Page 8, line 29: Do not abbreviate 'Formation'.

36. Right.

36. Changed.

37. Page 8, line 34: The river name should be 'paleo-Seymareh river' as this is not an officially recognized name.

37. Right.

37. Changed.

38. Page 9, line 12: 'literature' is insufficient. What were the sources?

38. Right.

38. **"The geomorphological study of the area was carried out first through the analysis and interpretation of remote sensing data, such as aerial photos (National Cartographic Center of Iran, aerial photo, scale:1:20000, acquired on 24 August 1955), Google Earth satellite optical images (2018 Landsat Imagery) and vector topographic maps (National Cartographic Center of Iran, topographic map of Kuhdasht, scale: 1:25,000), which led to the first detection of possible geomorphic markers within the Seymareh River valley. Vector topographic data also allowed the construction of a 10 m Digital Elevation Model (DEM) for terrain analyses and led to the projection of the possible geomorphic markers along the river longitudinal profile (Wilson and Gallant; 2001; Burbank and Anderson, 2012).**

**The DEM was obtained by the ArcGIS 10® software package, starting from vector topographic data (contour lines, hydrography and point elevation) and using the ANUDEM interpolation algorithm (Hutchinson et al., 2011 and references therein)."**

39. Page 9, line 18: Unclear what is meant by the 'data where [sic] acquired'.

39. Right.

39. Changed.

40. Page 9, line 20: Write 'minutes' out in full.

40. Right.

40. Changed.

41. Page 10, line 4: Which components of the Hydrology toolbox were used?

41. Right.

41. **"To automatically extract the hydrographic network from the DEM and then to project the geomorphic markers along the longitudinal river profiles, some of the ArcGIS® 10 tools of the Hydrology toolbox were used (Flow Direction, Flow Accumulation, Reclassify, Stream Order and Stream to Feature), setting the flow accumulation threshold according to that proposed for the fluvial domain ($10^{-1}$ km$^2$) by Montgomery and Foufoula-Georgiu (1993). The longitudinal profile was therefore transformed into a route along which the elevation of the top surfaces of geomorphic markers identified in the area were projected through the Linear Referencing Tools (Create Route and Locate Features along Route)."**

42. Page 10, line 17: Should be 'the Seymareh River gorge'.

42. Right.

42. Changed.

43. Page 11, line 10: Why is this knickpoint '. . .the most interesting...'? The authors seem to be implying that this may related to instability within the flank of Kabir Kuh, but the reader can only guess.

43. Right.

43. Changed.

44. Page 13, line 20: Lake Balmak is mentioned here for the first time. Why?

44. Right.

44. Changed.

45. Page 13, line 24: What about the drainage was progressive? The actual drainage is now well characterized here.

44. Right.

44. Changed.

46. Page 13, line 28: This duration for the lake needs to be compared with estimates provided in other sources.

46. We agree with concept of comparing the results with other sources. In this regard, we compare the result for the persistence of the Seymareh lake of about 5 ky with the estimate of 935 years supposed on the same lake by Shoaei (2014).

46. **As indicated by the age of the Qt1_m terrace (of 4.49±0.48 ka), the Seymareh Lake likely persisted up to ~5 ka,**

10  **much longer than the 935 years estimated by Shoaei (2014)".**

47. Page 13, line 30: This is the only place this figure is cited. The figure does not alone indicate what the authors suggest. Has some further work been done on this stratigraphic section that I missed?

47. Right.

15  47. Changed.

48. Page 14, line 3: What is meant by 'time scan'?

48. Right.

48. Changed.

49. Page 14, line 5: I again see no benefit to a list instead of writing out the description in proper paragraph format.

49. Right.

49. Changed.

25  50. Text in some figures, particularly the labels (and markers) for sample locations in Figs. 7 and 10, is too small and thus hardly legible.

50. Right.

50. Changed.

**REV#2: (D. Petley)**

**FUNDAMENTAL ISSUES:**

1. The manuscript is very poorly organised. It is hard to understand what is new, what is a reinterpretation, and what is background information. I found the paper difficult to read and to follow, and at the end I am not sure I really managed to work out what was new.

1. We absolutely agree.

1. We re-organized the manuscript balancing the content between background information and new data. In particular:

a) we rewrote the introduction to better focus the gap of knowledge we want to fill (lack of time constraints to the Seymareh River valley evolution before and after the Seymareh Landslide occurrence, to outline the role of the geomorphic processes both as predisposing factors for MRC processes and as response to this giant gravitational instability;

b) we moved any new observation to the Results and any interpretation to the Discussion, which was organized in 4 subsections to improve the readability.

(SEE ALSO THE RESPONSES to REV#1)

2. There is a huge amount of background information. Much of this seems to be irrelevant or tangential. In some cases, it misrepresents the literature (e.g to say "which is mainly focused on predictive models" in page 2 is not correct. The authors really need to work out what is needed and what is not.

2. We agree.

2. We reduced the part concerning the background information by 50% into a unique section: "**2 Regional geological and geomorphological framework**"., focusing on what of the literature is needed for the present study. We also moved the paragraph relative to Seismicity and to the general Tectonics of the area to the supplementary material. Furthermore, the section on the "Revised stratigraphic column and geological sections of the Seymareh river valley" was removed.

(SEE ALSO THE RESPONSES to REV#1)

3. Very little of the paper is really about predisposing factors. This seems to be focused on knickpoint migration, but it is it clear as to whether this is really a factor in such a large landslide.

3. We agree.

3.We refocused the topic of the paper highlighting the real new insights on this exceptional case of study: new time constraints to the Seymareh River valley evolution before and after the emplacement of the Seymareh landslide, to outline the role of the geomorphic processes both as predisposing factors for MRC processes and as response to this giant gravitational instability. We then discussed the "**Implications of the evolutionary model for future back-analysis of the SL**" in a subsection of the Discussion. Accordingly, we changed also the Title of the manuscript into **"New constraints to**

river valley evolution before and after the emplacement of the largest landslide on the exposed Earth surface: the Seymareh rockslide - debris avalanche (Zagros Mts., Iran)".

Attached the new **"5.4 Implications of the evolutionary model for future back-analysis of the SL**" subsection:

"According to the multi-modeling approach proposed by Martino et al. (2017), Quaternary landscape evolution modeling of slope-to-valley floor systems plays a key role as a tool for chronological constraints to the creep evolution of entire slopes (Bozzano et al., 2016; Della Seta et al., 2017).

The geomorphic processes developed before the failure of the SL likely acted as predisposing factors for MRC processes in the rock mass successively collapsed. Kinematic freedom, both at the top and on the fold flank was created by the incising network of streams that dissect the Asmari Formation carbonate caprock following the major joint set in the Asmari Formation already described in Roberts and Evans (2013; and references therein). In particular, the headward erosion of streams towards the anticline's structural high described by Oberlander (1968), caused the expansion of the fold axial basins through the softer units, determining the upslope kinematic freedom. In the timing proposed by Tucker and Slingerland (1996) the latter was reached at approximately 1.6 ka. Stress release at the slope base was definitely produced by the Middle-Late Pleistocene upstream migration of the knickpoint along the Seymareh River longitudinal profile. Unfortunately, since the emplacement of the landslide swept away the uppermost outcrops of the alluvial terraces formed in response to the upstream knickpoint migration, the rate of knickpoint migration cannot be inferred. Nonetheless, an elapsed time-to-failure on the order of $10^2$ ky, since the kinematic freedom at the slope base was reached, can be reasonably estimated by the age of the oldest terrace in the lower reach of the river minus the age of the landslide occurrence.

It is noteworthy that the stratigraphy of the source rock mass, also described in detail by Roberts and Evans (2013), accounts for different rheological behaviors, which could have induced differential strain rates within the slope leading to failure according to a MRC process. More particularly, the time-dependent visco-plastic behavior, more typical of clayey and marly deposits, which have lower viscosity values, can justify time-dependent (creep) strains which could have generated high stress concentration within the higher viscosity level over time (i.e., mostly characterized by elasto-plastic rheology), inducing their cracking and leading to failure. In fact, a stiffness contrast exists between the upper member of the Pabdeh Formation and the overlying Asmari Formation. The attitude of the strata is moderately dipping downslope (15°-20°), and a reduced lateral confining effect is due to continental and epicontinental deposits ascribable to the Gachsaran and Agha Jari Formations. Moreover, the low dip angle of the strata reduces the vertical thickness of the Asmari Formation caprock, which was completely eroded by Seymareh River during its engraving, thus allowing the sliding mechanism of the Pabdeh and Asmari layered formations.

Therefore, the results of this work have implications for a future back-analysis through stress-strain numerical modeling of the SL because they can be used to constrain the elapsed time since MRC initiation and ultimate failure conditions. Such a perspective is to be regarded as a key challenge for dimensioning such an end member event in

regard to both time and space distribution as well as for evaluating the possible role of impulsive triggering actions (i.e., strong to very strong earthquakes) in anticipating the time-to-failure of the slope."

4. The sections on the post-event landscape evolution is interesting, and probably represents the best part of the paper. But it needs to be organised in a more systematic manner that allows the reader to follow the argument. At the moment it feels somewhat chaotic and disorganised, and extremely difficult to follow. The substantive part of this section (5.2) is brief and hard to follow. There are results elsewhere though, which is confusing.

4. We agree.

4. We re-organized the results, formatting a new more concise results section without interpretations. Then we re-organized the Discussion in 4 subsections to improve the readability.

Attached the new **"4 Results"** section:

[revised manuscript text omitted]

5. I am not sure why so much detail is needed on the large-scale geomorphic evolution of the area. This would be better dealt with through references.

5. We agree

5. We reduced the text on the geomorphic evolution into a unique section: "**2 Regional geological and geomorphological framework**"., focusing on what of the literature is needed for the present study.

Attached the paragraph regarding the large-scale geomorphic evolution of the area:

**"The Zagros Range globally provides one of the most spectacular examples of landscape evolution in response to active tectonics (Bourne and Twidale, 2011) because its drainage network clearly adapted to the growth of the thrust-fold structures (Ramsey et al., 2008) and to the erodibility of the outcropping formations (Oberlander, 1985). Oberlander (1968) suggested that the drainage network in the NW Zagros was superimposed from structurally conformable younger horizons. In his model, the breaching of hard geological units of the antiformal ridges follows a phase of river cutting and expansion of the fold axial fold basins through the softer overlying units. In the Kabir-kuh fold, the transverse cutting of the Asmari limestone, and the exposure of the underlying more erodible Pabdeh-Gurpi marls, leads to the formation of a low-relief landscape with synformal ridges on which the new through-going drainage system can be developed. In Oberlander's hypothesis, it is the Pabdeh and Gurpi marls that facilitate the creation of a low-relief landscape across the anticline crests and are therefore integral to the story of drainage superimposition.**

**Tucker and Slingerland (1996) computed a numerical landscape evolution model, calibrated on the Kabir-kuh fold, to understand how the growth and propagation of the folds, the different lithologies and the drainage network can influence the sediment flux from a tectonically active belt towards the foreland basin. The authors calibrated the landscape evolution model with the current topography of the range, obtaining time constraints for landscape evolution modeling. According to the Oberlander model, Fig. 3 shows four main steps that describe the landscape evolution of the Kabir-kuh fold with the timing provided in the model by Tucker and Slingerland (1996).**

**Step 1 - Approximately 4.3 Ma, in response to the initial stages of fold growth, an orthoclinal drainage develops, parallel to the main structures. The tributaries flowing along the flanks of the folds transport debris, which is deposited in the synclines. In the Kabir-kuh fold the carbonate core is still buried by the Miocene cover units.**

**Step 2 - Approximately 3.8 Ma, as soon as the deformation front migrates towards the SW, new folds raise with a progressive adjustment of the drainage to these morpho-structures. The previously deposited sediments are remobilized and transported towards the depocenter of the syncline basins and partly outside; the syn-orogenic**

deposits are strongly eroded along the crests of the anticlines, thus exposing the underlying formations. This causes a topography characterized by resistant hogbacks that flank the inner cores.

Step 3 - Approximately 2.4 Ma, with the ongoing deformation, the drainage develops in a "trellis" pattern. The river erosion affects the erodible units located stratigraphically between the limestone of the Asmari Formation and the inner core of the fold. At the end of this step the Miocene cover is completely removed from the ridges and the river erosion also affects the marls and evaporites of the syn-orogenic formations in the valleys, exposing the underlying limestone of the Asmari Formation.

Step 4 - Approximately 1.6 Ma, due to the continuous uplift and exhumation of younger, more external folds, the sediment accumulation becomes negligible and the Asmari limestone is strongly eroded giving rise to syncline ridges. The following Quaternary landscape evolution is then likely driven by the evolution of the drainage network and is also influenced by climatic factors and by the slope-to-channel dynamics.

The model by Tucker and Slingerland (1996) is the unique numerical model existing on the Kabir-kuh fold and this motivates our choice of using it as a reference for the medium-to-long term evolution of the Seymareh River valley.

The Seymareh River valley is arranged parallel to the Kabir-kuh fold and its evolution was inevitably influenced by the exceptional landslide event that temporarily dammed it, causing the formation of the three-lake system which includes the Seymareh, Jaidar and Balmak lakes (Fig. 4). The valley evolution before and after the event is well recorded by Quaternary landforms preserved along the valley. Yamani et al. (2012) focused on the post-failure evolution of the valley describing four levels of terraces upstream of the landslide dams as a sequence of lacustrine terraces. Shoaei (2014), in addition to evaluating the longevity of the SL dams, identified in the merging of Seymareh River with a left tributary as the reason for strong river incision at the base of the northeastern flank of the Kabir-kuh fold and as a possible causal factor for the SL collapse.

However, none of the previous studies on the Quaternary evolution of the Seymareh River valley provided absolute dating of

geomorphic markers (mainly fluvial terraces) preserved upstream as well as downstream of the landslide dam or provided robust and quantitative constraints to the pre-failure valley evolution as a possible geomorphological factor for failure occurrence."

6. The discussion is also hard to follow, needing a restructure.

6. We agree.

6. We re-organized the discussion section, formatting a new one as follows:

[revised manuscript text omitted]

7. I am not sure that the review of previous studies of this landslide really present them in a correct manner. I think the abstract needs rewriting - it does not present the contents of the paper well.

7. We agree.

7. The summary of previous work on the Seymareh landslide was corrected as following:

**“Different interpretations have been proposed so far by the scientific community to explain the generation of such an exceptional event and different scenarios have been hypothesized for explaining the induced changes in landscape. Harrison and Falcon (1937, 1938) provided much of the present knowledge on the rock avalanche, including the geology and structure of the source area, the general geomorphology and the basic geometry of the landslide. Oberlander (1965) included a short appendix on the landslide in his study of Zagros streams and discussed its origin in relation to the activity of Seymareh River. Later, in the 1960s, Watson and Wright (1969) characterized the geomorphology and stratigraphy of the debris, discussed the origin of the initial rockslide, and examined the debris avalanche emplacement mechanisms. Roberts (2008) and Roberts and Evans (2013) provided a detailed model of how the geological and tectonic evolution of the Kabir-kuh fold predisposed the slope to such a large-scale failure, including formation of structural/kinematic and rheological control, and inferred a seismic trigger. Specifically, Roberts and Evans (2013) obtained from a charcoal-rich layer approximately 15 m above the base of the lacustrine sequence with a [14]C age of 8710 years BP. Based on the interpretation of three separate radiocarbon ages provided additionally by Griffiths et al. (2001) an estimated radiocarbon bracket age of the Seymareh event was suggested**

between 9800–8710 [14]C years BP. Yamani et al. (2012) provided some general details on the evolution of the dam lake drainage, describing a sequence of entrenched lacustrine terraces upstream of the landslide dam. Finally, Shoaei (2014) reviewed the possible mechanisms of failure and interpreted the post-failure geomorphic features through analyzing the processes responsible for the formation and erosion of the landslide dams of the Seymareh, Jaidar and Balmak (called also Chah Javal) lakes by using available annual sedimentation data and field measurements of the deposits in these lakes."

The abstract was rewritten as following:

"The Seymareh Landslide detached ~10 ka from the northeastern flank of the Kabir-kuh fold (Zagros Mts., Iran), is recognized worldwide as the largest rock slope failure (44 Gm$^3$) ever recorded on the exposed Earth surface. Detailed studies have been performed that have described the landslide mechanism and different scenarios have been proposed for explaining the induced changes in landscape. The purpose of this study is to provide still missing time constraints to the evolution of the Seymareh River valley, before and after the emplacement of the Seymareh Landslide, to highlight the role of geomorphic processes both as predisposing factors and as response to the landslide debris emplacement.

We used optically stimulated luminescence (OSL) to date lacustrine and fluvial terrace sediments, whose plano-altimetric distribution has been correlated to the detectable knickpoints along the Seymareh River longitudinal profile, allowing the reconstruction of the evolutionary model of the fluvial valley. We infer that the knickpoint migration along the main river and the erosion wave propagated upstream through the whole drainage network caused the stress release and the ultimate failure of the rock mass involved in the landslide. We estimated that the stress release activated a Mass Rock Creep (MRC) process with gravity-driven deformation processes occurring over an elapsed time-to-failure on the order of $10^2$ ky. We estimated also that the Seymareh damming lake persisted for ~3500 years before starting to empty ~6.6 ka due to lake overflow. A sedimentation rate of 10 mm y$^{-1}$ was estimated for the lacustrine deposits, which increased up to 17 mm y$^{-1}$ during the early stage of lake emptying due to the increased sediment yield from the lake tributaries. We calculated an erosion rate of 1.8 cm y$^{-1}$ since the beginning of the landslide cut by Seymareh River, which propagated through the drainage system up to the landslide source area. The evolutionary model of the Seymareh River valley can provide the necessary constraints for future stress-strain numerical modeling of the landslide slope to reproduce the MRC and demonstrate the possible role of seismic forcing in anticipating the time-to-failure for such an end-member case study."

8. I also recommend that the authors think carefully about the figures. Fig 5 for example dows not seem to really present the information being presented, figre 16 is impossible to understand, Figure 11 needs annotation, Fig 7 is too complex to understand in its current form.

8. We agree.

8. We revised Fig. 5 (new Fig. 2), removing the section C-C' and including the photo of the outcropping stratigraphy of Fig. 13. Fig. 16 (new Fig. 12) was implemented information for a better interpretation. Fig. 7 (new Fig. 6) was improved increasing the text size. We do not understand what kind of annotation is needed in Fig. 11.

[Figure]

**Figure 2: Geological Map, stratigraphic column and cross sections of the study area.**

[Figure]

**Figure 6: Map of the lacustrine and alluvial terrace suite and of the most significant landforms for the valley slope evolution upstream of the SL.**

[Figure]

**Figure 12: Evolutionary model of the Seymareh River valley. See text for explanation. Traces and legend of geological cross-sections are reported in Fig. 2.**

**AMERICAN JOURNAL EXPERTS**

**EDITORIAL CERTIFICATE**

This document certifies that the manuscript listed below was edited for proper English language, grammar, punctuation, spelling, and overall style by one or more of the highly qualified native English speaking editors at American Journal Experts.

**Manuscript title:**

New constraints to river valley evolution before and after the emplacement of the largest landslide on emerged Earth surface: the Seymareh rock slide - debris avalanche (Zagros Mts., Iran)

**Authors:**

Michele Delchiaro, Marta Della Seta, Salvatore Martino, Maryam Dehbozorgi, Reza Nozaem

**Date Issued:**

April 24, 2019

**Certificate Verification Key:**

A83A-E037-B9A8-6291-476D

[Figure]

This certificate may be verified at www.aje.com/certificate. This document certifies that the manuscript listed above was edited for proper English language, grammar, punctuation, spelling, and overall style by one or more of the highly qualified native English speaking editors at American Journal Experts. Neither the research content nor the authors' intentions were altered in any way during the editing process. Documents receiving this certification should be English-ready for publication; however, the author has the ability to accept or reject our suggestions and changes. To verify the final AJE edited version, please visit our verification page. If you have any questions or concerns about this edited document, please contact American Journal Experts at support@aje.com.

American Journal Experts provides a range of editing, translation and manuscript services for researchers and publishers around the world. Our top-quality PhD editors are all native English speakers from America's top universities. Our editors come from nearly every research field and possess the highest qualifications to edit research manuscripts written by non-native English speakers. For more information about our company, services and partner discounts, please visit www.aje.com.

---

## Author Response (AR2)

**in black reviewer comments**

**in red our response**

**in green changes in the manuscript**

**REV#1: (N. Roberts)**

The manuscript is a re-submission of an earlier version that required very extensive modification to enable clear communication of the conducted study. The authors have greatly improved this updated version, with the most notable changes including:

1. Reorganization to follow a logical flow.
2. Removing superfluous material.
3. Improving the writing for correctness and clarity, although numerous minor errors remain.
4. Streamlining of display items, although several require further improvement for legibility.

With the major issues from the previous submission addressed, some of the finer points of the paper now come through. This also highlights a large number of relatively minor issues, comprising mainly unclear/ambiguous statements or remaining problems with the writing. A number of one-sentence paragraphs also remain. Several figures require additional modification to ensure their legibility.

The site descriptions (of geomorphic markers and OSL dating) stand out as needing further attention. The authors provide only very minimal geomorphic description of the valley-bottom features they study. Unfortunately, stratigraphic and sedimentologic details are almost completely absent. Both the morphology and composition of the landforms/deposits pertinent to the study need to be better, although concisely, characterized.

A large number of minor edits will need to be made. The reference list should be carefully checked against citations in the text. The reference for Elyasi et al. (2014), which is cited on page 4, is missing. Others may be missing as well.

This revised manuscript can be accepted following some further, minor revision outlined below.

**TECHNICAL ISSUES (some of which relate to odd or ambiguous wording)**

1. Title: Does the title really need to be this long? Also note that this landslide is a 'rock avalanche' not a 'debris avalanche' (see also comment below for page 2, line 20).

1. We agree.

1. We changed the title into "**Reconstruction of the river valley evolution before and after the emplacement of the giant Seymareh rock avalanche (Zagros Mts., Iran)**".

2. Page 1, line 25: I don't understand the wording '…since the beginning of the landslide cut by Seymareh River…'. Is this supposed to mean the initiation of dam breaching?

2. We agree.

2. we changed the text into "…**the initiation of dam breaching by Seymareh River**…".

3. Page 1, line 26: The main post-landslide erosion migrated to the toe of the source slope. Some erosion has certainly occurred in the weaker units of the sources are, but his is on a different scale than that experienced in the valley bottom. Ideally, the wording should reflect this difference, and could even not the different erosion patters (i.e. deep entrenchment of the landslide debris and associated lake deposits in the valley bottom, compared to a shallow, well integrated drainage network in the weakest lithologies of the source area).

3. We do not completely agree. Indeed, the erosive capacity of a drainage system within a basin is strongly conditioned by changes in the base level. In this regard, the sharp drop in the base level following the dam breaching affected the entire drainage network (including the one developed on the weakest lithologies of the scar area) increasing its erosive capacity.

3. We preferred to leave the wording unchanged, avoiding speculating on topics not covered by this study.

4. Page 1, line 28: The text '…the possible role of seismic forcing in anticipating the time-to-failure…' is confusing. I presume the authors mean the role of seismic triggering in prematurely terminating the creep-controlled time-to-failure pathway. If this argument is being made in the paper, consider citing Hermanns and Longva (2012) - as mentioned in the review of the initial manuscript - as they discuss just such a situation (see their Figure 6.1). After reading the rest of the manuscript, however, I see that this topic is really not fleshed out. In that case, should this topic even be part of the abstract?

4. We partially agree. Undoubtedly, the seismic trigger of the Seymareh landslide is not the subject of this study;

nonetheless, the results of this study have considerable implications for further researches in this perspective.

4. We left the wording unchanged for the aforementioned reasons.

5. Page 1, line 26: Why is this an 'end-member case study'? Presumably this is because the landslide was so large. Clarify.

5. We agree

5. we changed the sentence into "… **for such an extremely large case study**…".

6. Page 2, line 9: The shift from linearly increasing to nonlinearly increasing erosion needs to be clarified. Presumably the authors mean that the nonlinear increase is a response to perturbation of valley-bottom erosion by bedrock landslides, but this is unclear. The nature of the rate of increase – accelerating or deceleration – is also unclear. This would of course be complex as impoundment of lakes will lead to short-lived deposition and storage of sediment, but their later drainage due to dam failure will result in short-lived increases in erosion.

6. The shift from linear to non-linear increase of erosion is caused by the shift from prevailing river erosion to landslide-related erosion caused by the reaching of threshold slope conditions associated with the hillslope material strength. This statement refers to regional/long-term changes in landscapes and not to the possible local/short-term perturbations the reviewer refers to at the end of this comment.

6. We rephrased as follows: "**In response to rock uplift, relief and hillslope angles increase linearly in time mainly due to fluvial erosion processes in landscapes affected by low to moderate tectonic forcing (Montgomery and Brandon, 2002; Binnie et al., 2007; Larsen and Montgomery, 2012). Nonetheless, such a linear increase in relief and hillslope angles is limited by the reaching of the threshold slope conditions associated with the hillslope material strength (Schmidt and Montgomery, 1995), until the latter is exceeded by gravitational stress giving rise to bedrock landslides. This leads to a nonlinear increase in erosion rates in landscapes affected by long-lasting or high-rate tectonic forcing, where the increase in the rate of channel incision is accommodated by an increased frequency of slope failure rather than by slope steepening.**"

7. Page 2, line 11: What is meant by 'the ultimate conditions of rock mass creep'? Do the authors mean that rock mass creep culminates in slope failure?

7. Yes, we mean that mass rock creep process can culminate in slope failure when the tertiary creep stage is reached.

7. We rephrased into "…. **the terminal phase of mass rock creep**…".

8. Page 2, first paragraph: The introduction of rock mass creep is missing key references to the different stages of creep (i.e. mechanisms of stationary creep and accelerating creep), such as work by Saito and later on that of Petley and colleagues. Citing a couple of these, particularly those directly applicable to bedrock landslides, would be very helpful to the reader.

8. We agree.

8. We added the following references:

"**Petley, D. N. and Allison, R. J. (1997): The mechanics of deep-seated landslides. Earth Surface Processes and Landforms: The Journal of the British Geomorphological Group, 22(8), 747-758.**
**Saito, M. (1969): Forecasting time of slope failure by tertiary creep. In Proceedings of the 7th International Conference on Soil Mechanics and Foundation Engineering, Mexico City, Vol. 2, pp. 677–683**."

9. Page 2, line 13: What is meant by 'generalized failure'?

9. We mean that the failure involved suddenly the entire slope.

9. We rephrased as "… **sudden and complete slope collapse**...".

10. Page 2, line 17: There are two issues with the details in this sentence. Firstly, rock mass creep does not lead directly to rock avalanches, but rather to preliminary transport mechanisms, largely sliding. Large rock slides, if able to accelerate and run out, then transform into rock avalanches. Secondly, although creep may fragment the rock mass some prior to catastrophic failure, rock mass fragmentation largely occurs during transport (see Davies and McSaveney, 2009. The role of rock fragmentation in the motion of large landslides: Eng. Geol. 109). In either case, the fragmentation is not instantaneous as the authors have suggested.

10. We agree.

10. "**Catastrophic failures occurring in the accelerating MRC stage can evolve rapidly into rock avalanches due to the fragmentation of rock masses during their transport (Hungr et al., 2001; Davies and McSaveney, 2009).**"

11. Page 2, line 20: The second phase of failure of the Seymareh landslide was a rock avalanche, not a debris avalanche (this also needs to be corrected in the title). See Hungr et al. (2001, cited in this sentence of the manuscript) and further details in Hungr et al. (2014. The Varnes classification of landslide types, an update. Landslides 11). Why is the material type specified in 'largest subaerial rock landslide'? I am unaware of any subaerial soil/sediments failures larger than this and, unless the authors know of one, 'rock' can be dropped. A citation would be good to support the statement – probably Roberts

and Evans (2013) since the paper provides the more recent (and accurate) volume estimate for the landslide and demonstrates in its supplementary material that is it larger than the Baga Bogd rockslide, Mongolia.

11. We agree.

11. We rephrased as "**This work is focused on the evolution of the Seymareh River valley before and after the valley was dammed by the landslide worldwide recognized as the largest subaerial landslide ever observed, the Seymareh rock avalanche (Roberts and Evans, 2013) that occurred in the northwestern Zagros Mts. (Iran).**"

12. Page 3, line 16: Consider dropping '…the emplacement of…' as this study pertains to valley-bottom evolution due to causation of the landslide as well as subsequent debris emplacement.

12. For sure, we reconstruct the evolutionary model of the Seymareh River valley before and after the landslide "occurrence".

12. We dropped '…**the emplacement of**…'.

13. Page 3, line 17: Is it really necessary to abbreviate 'Seymareh landslide' to 'SL'? Also, capitalization should be 'the Seymareh landslide' as this isn't a truly formal name.

13. We referred to Shoaei (2014), who named the Seymareh Landslide "SL". Nonetheless, it is not necessary.

13. We changed "**SL**" into "**Seymareh landslide**" throughout the manuscript.

14. Page 5: This is a nice summary of Tucker and Singerland's (1996) landscape evolution model. However, it is probably a bit detailed for the background section. Considering moving the current, detailed description to the supplementary material and presenting a summary only in section 2. The authors might consider noting very briefly how the chronostratigraphy reported by Homke et al. (2004) agrees or disagrees with this model.

14. In our opinion the summary of Tucker and Slingerland's (1996) LEM is functional for commenting Fig.3 and, consequently, for justifying our reference to this model for the medium-to-long term evolution of the Seymareh River valley. Furthermore, we did not address the request of noting very briefly how the chronostratigraphy reported by Homke et al. (2004) agrees or disagrees with this model, since the T&S's model was implemented to be representative of the landscape evolution of the whole Simply Folded Zagros, while the Chronostratigraphy by Homke et al. is site-specifically referred to the Changuleh fold. In our opinion the two works are not comparable.

14. We left the wording unchanged

15. Page 6, line 4: As these lakes no longer exist, use 'included' not 'includes'.

15. We agree.

15. Done

16. Page 4, line 8: The meaning of '…identified in the merging of Seymareh River with a left tributary as the reason…' is unclear. Naming the tributary would provide further useful context.

16. We agree.

16. "….**in the merging of Seymareh River with the Kashgan River left tributary as the reason**….".

17. Page 7, line 4: What is meant by '…state of thickening…'?

17. We mean the compaction degree.

17. We removed it.

15  18. Page 7, line 4: In identifying locations for OSL sampling, what consideration was giving to selecting sediments that had the highest likelihood to have been fully bleached during transport? Could Seymareh River's high suspended sediment load prevented full bleaching of sediment? Or is the assumption that any sediment making its way into the river is was fully bleached during slope-wash transport to the river or one of its tributaries? Is not addressed here, this should be noted in the discussion. Sample locations provided in many of the figures are very hard to make out due to the figures' small scale.

20  18. The exposure to the light of the samples could cause the reducing or the total bleaching of the luminescence signal. Therefore, the sampling was carried out according to a standard procedure (described in detail in the Methods section), which clearly indicates to take samples from parts of the deposit lying at least 1 m below the top surface or below eventual erosional surfaces recognized within the deposit. In this way one can prevent the signal to be reset. The limits of the OSL method have been evidenced both in the results and in the discussion (Constraints to pre-failure valley evolution ) in which
25  minimum ages of 373±34 ka and 312±45 ka for samples SEY3 and SEY10 are presented, respectively, since these samples were saturated due to their low concentration of quartz grains.

As for the low resolution of sample locations in figures, we agree. Nonetheless, in the Supplementary Material we provided the single forms for each sampling sites, along with coordinates, photos and main sedimentological characters of the sampled deposits.

18. We enlarged the scale of figures.

19. Page 7, line 9: The description sounds like the tube may have been inserted vertically into the ground. The typically protocol is to insert the tube horizontally into a vertical face. Clarify what was done.

19. The tube was inserted horizontally, as you can see also by the photos in the Supplementary Material.

19. "…. **horizontally into a vertical face**….".

20. Page 7, line 10: The description of the 30 cm of sediment surrounding the tube is confusing. Does this mean the sampling locations comprised zones of homogeneous sediment wide and thick enough that the tube could be inserted with an at least 30-cm zone of in situ sediment remaining around it? Or is this meant to indicate that when the sample was removed a zone of 30 cm of sediment around it was somehow retained?

20. We mean that the sampling locations comprised zones of homogeneous sediment wide and thick at least 30 cm.

20. We rephrased as "…**the tube was inserted into zones of homogeneous sediment at least 30 cm wide and thick.**"

21. Page 7, line 22: What is the 'SAR protocol'? This is neither explained in the text nor supported with a specification citation. Presumably SAR is an abbreviation, and if so will need to be defined.

21. We agree.

21. we added the following explanation: "…. **single-aliquot regenerative-dose (SAR) protocol (Murray and Olley, 2002; Wintle and Murray, 2006)** ….".

22. Page 8, line 1: The sentence starting on this line is quite hard to follow. Consider breaking it up based on the two elements it describes: the sediment compositions of the terraces, and relationship of the terraces to landforms (i.e. inactive fans). Also, stating the sediment consist 'mainly' of every possible sediment size is not helpful. Persuadably the lacustrine deposits are finer the alluvial ones, but this is not clear from the description. What types of sedimentary structures are present? What types of a depositional environment is suggested by the alluvial deposits – a high-energy braided system or a low-energy meandering system?

22. We agree about breaking up the sentence. On the other hand, detailed description (including sedimentary structures, depositional environment ecc.) of each sampled deposit had been already provided with the supplementary material.

22. We rephrased into "…. **Fluvial terraces consist of gravel, sand, silt and clay, in comparison to the lacustrine deposits mainly composed by silt (detailed information on each sampled deposit is provided with the supplementary material). Conglomerates outcropping immediately upstream and downstream of the landslide pertain to inactive alluvial fans connected to a relict position of the valley floor, likely of a paleo-Seymareh river.**"

23. Page 8, line 4: What is meant by the upstream markers in the midstream section? This is confusing given that the previous line mentions upstream markers (up-valley of the landslide) and downstream markers (down-valley of the landslide).

23. We agree. We corrected the typesetting mistake.

10   23. "…. **In the upper reach of the Seymareh River valley, the geomorphic markers include** …".

24. Page 8, line 10: What is meant by the natural damming being 'extended'?

24. We agree.

24. "…. **where Harrison and Falcon (1938), Roberts and Evans (2013) and Shoaei (2014) hypothesized this natural**
15   **damming lake could be located** ….".

25. Page 8, line 11: What evidence is there that these features were deposited during drainage of the lake as opposed prior to dam incision?

25. We agree.

20   25. We removed "…**which likely formed during the emptying phase of the lake…**"

26. Page 8, line 11: What is the basis for the interpretation of the timing of fan delta formation? Note also that this is an interpretation in the results section.

25. We agree.

25   25. We removed "…**which likely formed during the emptying phase of the lake…**"

27. Page 8, line 15: 'Coherent' is not the right word here. Presumably the point is that an older alluvial terrain underlying the

lake's fill sequence has been shown to predate the fill sequence. If I have correctly interpreted the implied stratigraphic relationship, then the relative ages are demonstrated by the stratigraphy along. The interesting contribution of the OSL ages here is that they quantify the time spans between deposition of the alluvial and failure of the landslide (ca. 7 ka) and between failure of the landslide and the late phase of the lake (ca. 3 ka).

27. We agree.

27.We rephrased into "….**The OSL age of 17.9±1.50 ka (SEY6) obtained for an alluvial deposit at the base of the lacustrine deposits predated by ca. 7 ky the emplacement of the Seymareh landslide, according also to the time constraints provided by Roberts and Evans (2013)**….".

28. Page 8, line 17: How is the flood plain 'shaped' onto the landslide. Presumably this is largely a depositional feature. Is erosion also involved?

28. Strath terrace formation testify to phases of lateral erosion by the river. We identified also a surface with evidence of river flooding.

28. We changed "**shaped**" into "**formed**".

29. Page 8, line 17: Stating that 'clear evidence of MRC' was observed is unconvincing unless these features are described here, including by citing the relevant figure.

29. We agree.

29. We rephrased into '**As shown in Fig. 11, evidence of MRC driving towards stress concentration**'.

30. Page 9, line 6: Black arrows are mentioned here, but since no figure is cited in this sentence, the location of these arrows is unclear.

30. We agree.

30. We added the citation to the Figure 9 and 10.

31. Page 9, line 14: Why is this sentence set aside as its own paragraph?

31. We agree.

31. We included it in the following paragraph.

32. Page 9, line 26: The relevant figure showing the knickpoint should be cited here.

32. We agree.

32. We added the citation of Figure 9.

33. Page 9, line 30: Explain why the base of the landslide debris sheet is important. Is it assumed to rest on the pre-failure river channel (or close to it)? If this contact is being used to suggest that the modern river profile passing through the relict landslide dam – at least in places – is coincident with the pre-failure river profile, the base of the landslide debris should be represented in Fig. 9. The bedrock exposure in Fig. 7 is not visible without the annotation as that part of the figure is small and completely in shadow. Is there a photo of the bedrock and its upper contact that would help convince the reader of this interpretation?

33. We agree. The base of the landslide debris as well the upstream contact between lacustrine deposits and bedrock as evidenced by benchmarks, are important because they suggest that actually Seymareh River is tracing its path before the landslide event.

33. We added the basal contact of the landslide in Figure 9 and also a photo representing the contact between the bedrock and the debris.

34. Page 10, line 22: The landslide volume was not estimated here. A citation is, therefore, required to attribute this estimate to the proper pervious study, especially since the volume is not quantified anywhere else in the manuscript. Come to think of it, why is that? Volume is a key metric of landslides, and is especially important in this case as the authors emphasize a multiple times that this study addresses a particularly large landslide.

34. We agree.

34. We added the citation.

35. Page 10, line 28: This discussion requires some detail on the textural composition of the landslide dam, but such details are missing here, despite being described in several previous geologic studies of the landslide. Internal drainage though the dam – or at least parts of it – is extremely likely given the coarse nature of the debris. It would be helpful to note the role of such drainage in other large damns (see the extensive literature on landslide dams, starting with the early work of John Costa

and Robert Schuster). The Usoi landslide dam is probably one of the best examples to consider given its large size (>1 Gm3) and relatively long duration of stability (in a historical context) due to the formation of stable internal drainage.

35 We agree

35 We added the sentence **"A large literature treated the effects of infiltration on the longevity of large landslide dams (Ischuk, 2011; Schuster and Alford, 2004; and references therein). Internal drainage through the Seymareh landslide dam – or at least parts of it – is extremely likely given the coarse nature of the debris (Watson and Wright, 1969; Roberts and Evans, 2013)."** before the sentence **"Nevertheless, the possible role of groundwater seepage within the pervious natural dam in balancing the Seymareh River discharge and delaying the dam overflow remains a questionable topic to be approached and solved in future studies."**

36. Page 10, line 32: It is unclear whether these 'fill terraces' are envisioned as fluvial terraces. This is just one example of the many instances where imprecise description of geomorphic and geologic features has weakened the author's interpretations and arguments.

36. In the geomorphological literature the subdivision among the fluvial terraces is between: fill terraces and strath terraces.

36. We added the word '**fluvial**' to '**fill terrace**' for a better comprehension.

37. Page 11, line 8: Is overly narrowly limited to this time? It seems possible that initial overflow could have predated this, and that the fluvial sculpting and deposition noted farther downstream along the dam could have lagged a bit behind, if the initial drainage was minor. Given the very limited geomorphic and geologic descriptions in section 4, it's hard to interpret such things.

37 We agree that initial overflow could have predated the formation of the first strath. Nonetheless, the dated strath is the oldest time-constrained marker of the dam breaching and since it is sculpted just few meters below the top of the landslide deposit (which is incoherent), its age can be reasonably used for the calculation of the incision rates after the overflow. We better specified this:

37 "**The strath terrace sculpted on the landslide deposit and dated at 6.59±0.49 ka is the oldest time-constrained marker of the natural dam breaching due to overflow, which caused the lake to empty**" and "**The oldest strath terrace sculpted onto the landslide deposit and dated at 6.59±0.49 ka is just few meters below its top. Therefore, it can be reasonably used to calculate the erosion rate affecting the landslide deposit after the overflow. The ratio between the thickness of the eroded sediment below the strath surface (~120 m) and the time elapsed since the beginning of the process (~ 6.59 ky) allows estimation of an erosion rate of 1.8 cm y-1 for Seymareh River along the gorge.**"

38. Section 5.3: Is a list really necessary here? I don't see any reason not to use proper paragraph structure.

38. We don't agree. In our opinion, in this case the list helps to better understand the sequence of events that took place in the valley before and after the emplacement of the landslide. Moreover, the list is associated with Figure 12 that shows the different time steps.

38. We left unchanged the list.

39. Page 12, line 25: Citations to relevant work are necessary when characterizing the typically creep behaviour of sepcific lithologies.

39. We agree.

39. We added the sentence "**To date, a quantification of such rheological properties is lacking for the lithological units of the Kabir-kuh fold. Nonetheless, some inferences can be done according to previous studies on rock masses affected by MRC (Apuani et al., 2007; Bozzano et al., 2012; Bretschneider et al., 2013; Della Seta et al., 2017)**."

40. Page 12, line 30: The reduced lateral confining effect needs to be briefly explained.

40. The confining effect of the epicontinental and continental formation, like Gachsaran and Agha Jari Formation, is related to their thickness upon the slope, which in turn depends on the dip angle of the strata below.

40. We rephrased into"…**The attitude of the strata is moderately dipping downslope (15°-20°), leading to a lower vertical thickness of (and consequently a reduced lateral confining effect by) the epicontinental and continental units, such as the Gachsaran and Agha Jari Formation.**".

41. Page 12, line 31: Clarify that the reduction in incision necessary is for kinematic freedom of the landslide is relative to a more steeply dipping sequence, which would require a greater depth of incision. Note that is also suggested based on consideration of this landslide by Roberts (2008, his Figure 4.1).

41. We agree.

4i. We rephrased into "…**Moreover, the low dip angle of the strata reduces also the vertical thickness of the Asmari Formation caprock (Roberts, 2008) and consequently the incision necessary for kinematic freedom of the landslide, relative to a more steeply dipping sequence, which would require a greater depth of incision.**"

42. Page 13, line 16: Was does the naming change here to 'the Seymareh giant landslide'?

42. We agree.

42. We decided to use "…**Seymareh landslide** …" throughout the manuscript.

**FIGURES**

1. Figure 3: Erroneous text in caption: '…to the Oberlander's model…'
1. We corrected into **"…according to the model by Oberlander (1985) and…"**

2. Figure 5: Consider locating photo positions in one of the map figures.
2. Done

3. Figure 6: Some elements are overly complex and thus hard to read. Can the 'gully' symbols be removed? They add a lot of visual complexity but are of little importance.
3. We simplified the symbol of gullies, improved the location of OSL sampling and adding the location of photos shown in Figs. 5 and 7.

4. Figure 7: Consider locating photo positions in one of the map figures. Can a clear photo of the bedrock outcropping below the landslide be added?
4. Done. We added a photo

5. Figure 8: Erroneous text in caption: '…the valley slopes evolution…'
5. Corrected.

6. Figures 9 & 10: Many features of the figure are far too small to be clearly legible. What is meant in the caption by '…upstream and downstream conglomerates; downstream fluvial terrace suite…'? The inset maps are too small and illegible until I zoom in quite far in the PDF. The label font with coloured fill and think black outlines (i.e. the terrace labels) are very hard to read because of the font/colouring used.
6. We deleted the features too small to be legible (the ones in the inset maps, that know are legible). We deleted '…**upstream and downstream conglomerates; downstream fluvial terrace suite…**' from the caption because it was not necessary. We left the symbols for terraces unchanged because in our opinion they can be understood thanks to the naming (Qt…) as well as their colors are the same used in the maps of Figs. 6 and 8. We represented, as requested, also the points where the basal contact of the landslide is exposed in the gorge.

7. Figure 11: Indicate the source of the imagery in panel A (including the actual satellite image, not just 'GoogleEarth'). Represent scale in panel B.
7. Done

8. Figure 12: Where is the legend for the colour fills? Even if this is a full-page figure, it will be hard to read.

7. The legend for color fills is the same as in maps and long profiles (Figs. 6, 8, 9, 10) and we specified this in the caption. Anyway, naming of the different terrace is specified in the figure. We also enlarged the text to make the figure more readable.

**LANGUAGE ISSUES (some examples only)**

1. Page 1, line 10: Capitalization should be 'The Seymareh landslide'. Alternatively, if threating this as a formalized geographic feature, it should be 'Seymareh Landslide…'.

10   1. We agree.

1. "**Seymareh landslide**".

2. Page 1, line 13: '…changes in landscape…' is incorrect English.

2. We agree.

15   2. "…. **landscape changes**…".

3. Page 2, line 7: '…by the reaching the threshold…'.

3. We agree.

3. "…. **the reaching of the threshold**…."

4. Page 3, line 1: '…evolution of the dam lake drainage…'. What is 'dam lake' meant to be? Dammed lake?

4. We agree.

4. "…. **Evolution of the dammed lake drainage**…."

25   5. Page 3, line 13: 'the Sichuan' is imprecise and confusing. This should be 'the Sichuan River basin'.

5. We agree.

5. "…. **Sichuan River basin**…."

6. Page 3, line 18: Placing the compound modifiers 'pre-failure and post-failure' after the terraces seems awkward. This sentence is quite long and hard to follow, so consider breaking it up.

6. We agree.

6. "**Detailed geomorphological mapping, correlation and dating of certain geomorphic markers (Burbank and Anderson, 2012) represented by pre- and post-failure fluvial and lacustrine terraces have been performed upstream and downstream of the landslide dam. We provide new time constraints to the Seymareh River valley evolution and outlining the role of the geomorphic processes both as predisposing factors for MRC processes and as response to this giant gravitational instability**."

7. Page 3, line 25: For clarity, consider '…originates convergence since Late Cretaceous time' as this is ongoing. Also, the current use of hyphens is confusing and, since 'Late Cretaceous' is an official part of the geologic timescale, incorrect. Also use of the backslash later in the line is confusing.

7. Simply a typesetting error.

7. We removed the hyphens.

NOTE: I have not listed issues beyond this point, but there are many that will need to be addressed.

We hope the new version of the manuscript, according to the above listed change addresses all the language issues.

[revised manuscript text omitted]